# Koopman Q-learning: Offline Reinforcement Learning via Symmetries of Dynamics

## Abstract

Offline reinforcement learning leverages large datasets to train policies without interactions with the environment. The learned policies may then be deployed in real-world settings where interactions are costly or dangerous. Current algorithms over-fit to the training dataset and as a consequence perform poorly when deployed to out-of-distribution generalizations of the environment. We aim to address these limitations by learning a Koopman latent representation which allows us to infer symmetries of the system's underlying dynamic. The latter is then utilized to extend the otherwise static offline dataset during training; this constitutes a novel data augmentation framework which reflects the system's dynamic and is thus to be interpreted as an exploration of the environments phase space. To obtain the symmetries we employ Koopman theory in which nonlinear dynamics are represented in terms of a linear operator acting on the space of measurement functions of the system and thus symmetries of the dynamics may be inferred directly. We provide novel theoretical results on the existence and nature of symmetries relevant for control systems such as reinforcement learning settings. Moreover, we empirically evaluate our method on several benchmark offline reinforcement learning tasks and datasets including D4RL, Metaworld and Robosuite and find that by using ourframework we consistently improve the state-of-the-art for Q-learning methods.

## 1 Introduction

The recent impressive advances in reinforcement learning (RL) range from robotics, to strategy games and recommendation systems (Kalashnikov et al., 2018; Li et al., 2010). Reinforcement learning is canonically regarded as an active learning process - also referred to as online RL - where the agent interacts with the environment at each training run. In contrast, offline RL algorithms learn from large, previously collected static datasets, and thus do not rely on environment interactions (Agarwal et al., 2020a; Ernst et al., 2005; Fujimoto et al., 2019). Online data collection is performed by simulations or by means of real world interactions e.g. robotics and in either scenario interactions maybe costly and/or dangerous.

In principle offline datasets only need to be collected once which alleviates the before-mentioned short-comings of costly online interactions. Offline datasets are typically collected using behavioral policies for the specific task ranging from, random policies, or near-optimal policies to human demonstrations. In particular, being able to leverage the latter is a major advantage of offline RL over online approaches, and then the learned policies can be deployed or finetuned on the desired environment. Offline RL has successfully been applied to learn agents that outperform the behavioral policy used to collect the data (Kumar et al., 2020; Wu et al., 2019; Agarwal et al., 2020b; Ernst et al., 2005). However algorithms admit major shortcomings in regard to over-fitting and overestimating the true state-action values of the distribution. One solution was recently propsed by Sinha et al. (2021), where they tested several data augmentation schemes to improve the performance and generalization capabilities of the learned policies.

However, despite the recent progress, learning from offline demonstrations is a tedious endeavour as the dataset typically does not cover the full state-action space. Moreover, offline RL algorithms per definition do not admit the possibility for further environment exploration to refine their distributions towards an optimal policy. It was argued previously that is basically impossible for an offline RL agent to learn an optimal policy as the generalization to near data generically leads to compounding

errors such as overestimation bias (Kumar et al., 2020). In this paper, we look at offline RL through the lens of Koopman spectral theory in which nonlinear dynamics are represented in terms of a linear operator acting on the space of measurement functions of the system. Through which the representation the symmetries of the dynamics may be inferred directly, and can then be used to guide data augmentation strategies see Figure 1. We further provide theoretical results on the existence on nature of symmetries relevant for control systems such as reinforcement learning. More specifically, we apply Koopman spectral theory by: first learning symmetries of the system's underlying dynamic in a self-supervised fashion from the static dataset, and second employing the latter to extend the offline dataset at training time by out-of-distribution values. As this reflects the system's dynamics the additional data is to be interpreted as an exploration of the environment's phase space.

Some prior works have explored symmetry of the state-action space in the context of Markov Decision Processes (MDP's) (Higgins et al., 2018; Balaraman & Andrew, 2004; van der Pol et al., 2020) since many control tasks exhibit apparent symmetries e.g. the classic cart-pole task which is symmetric across y-axis. However, the paradigm we introduce in this work is of a different nature entirely. The distinction is twofold: first, the symmetries are learned in a self-supervised way and are in general not apparent to the developer; second: we concern with symmetry transformation from state tuples $(s_t, s_{t+1}) \rightarrow (\tilde{s}_t, \tilde{s}_{t+1})$ which leave the action invariant inferred from the dynamics inherited by the behavioral policy of the underlying offline data. In other words we seek to derive a neighbourhood around a MDP tuple in the offline dataset in which the behavioral policy is likely to choose the same action based on its dynamics in the environment. In practice the Koopman latent space representation is learned in a self-supervised manner by training to predict the next state using a VAE model (Kingma & Welling, 2013).

To summarize, in this paper, we propose Koopman Forward (Conservative) Q-learning (KFC): a model-free Q-learning algorithm which uses the symmetries in the dynamics of the environment to guide data augmentation strategies. We also provide thorough theoretical justifications for KFC. Finally, we empirically test our approach on several challenging benchmark datasets from D4RL (Fu et al., 2021), MetaWorld (Yu et al., 2019) and Robosuite (Zhu et al., 2020) and find that by using KFC we can improve the state-of-the-art on most benchmark offline reinforcement learning tasks.

## 2 PRELIMINARIES AND BACKGROUND

### 2.1 OFFLINE RL & CONSERVATIVE Q-LEARNING

Reinforcement learning algorithms train policies to maximize the cumulative reward received by an agent who interacts with an environment. Formally the setting is given by a Markov decision process $(\mathcal{S}, \mathcal{A}, \rho, r, \gamma)$, with state space $\mathcal{S}$, action space $\mathcal{A}$, and $\rho(s_{t+1}|s_t, a_t)$ the transition density function from the current state and action to the next state. Moreover, $\gamma$ is the discount factor and $r(S_t)$ the reward function. At any discrete time the agent chooses an action $a_t \in \mathcal{A}$ according to its underlying policy $\pi_\theta(a_t|s_t)$ based on the information of the current state $s_t \in \mathcal{S}$ where the policy is parametrized by $\theta$. We focus on the Actor-Critic methods for continuous control tasks in the following. In deep RL the parameters $\theta$ are the weights in a deep neural network function approximation of the policy or Actor as well as the state-action value function $Q$ or Critic, respectively, and are optimized by gradient decent. The agent i.e. the Actor-Critic is trained to maximize the expected $\gamma$-discounted cumulative reward $\mathbb{E}_\pi[\sum_{t=0}^{T} \gamma^t r_\pi(s_t, a_t)]$, with respect to the policy network i.e. its parameters $\theta$. For notational simplicity we omit the explicit dependency of the latter in the remainder of this work. Furthermore the state-action value function $Q(s_t, a_t)$, returns the value of performing a given action $a_t$ while being in the state $s_t$. The Q-function is trained by minimizing the so called Bellman error as

$$\hat{Q}_{i+1} \leftarrow \arg\min_Q \mathbb{E}\left[ \left( r_t + \gamma \hat{Q}_i(s_{t+1}, a_{t+1}) - Q_i(s_t, a_t) \right)^2 \right] , \tag{1}$$

This is commonly referred to as the i[th] policy evaluation step where the hat denotes the target Q-function. In offline RL one aims to learn an optimal policy for the given the dataset $\mathcal{D} = \bigcup_{k=1,\dots,\#\text{data-points}} (s_t, a_t, r_t, s_{t+1})_k$ as the option for exploration of the MDP is not available. The policy is optimized to maximize the state-action value function via the policy improvement

$$\pi_{i+1} \leftarrow \arg\max_\pi \mathbb{E}_{s_t \sim \mathcal{D}}\left[ \hat{Q}(s_t, \pi_i(a_t|s_t)) \right] . \tag{2}$$

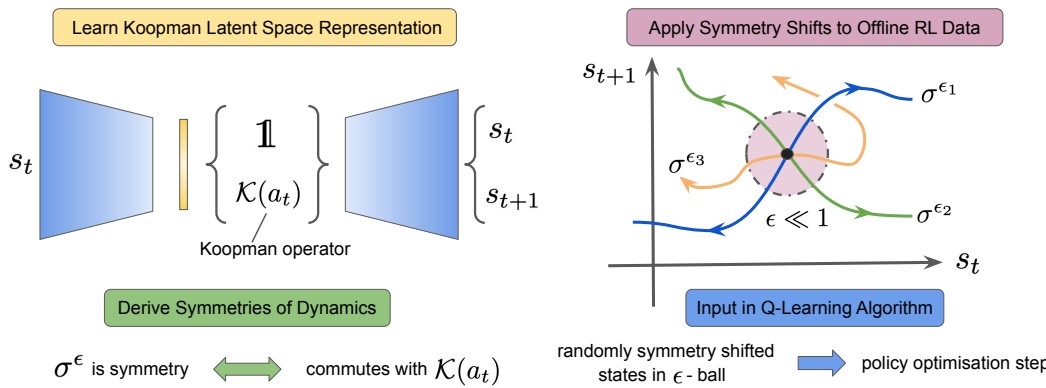

**Figure 1:** Overview of Koopman Q-learning. The states of the data point $(s_t, s_{t+1}, a_t)$ are shifted along symmetry trajectories parametrized by $\epsilon_{1,2,3}$ for constant $a_t$. Symmetry transformations can be combined to reach other specific subset of the $\epsilon$-ball region.

Note that behavioural policies including sub-optimal or randomized ones may be used to generate the static dataset $\mathcal{D}$. In that case offline RL algorithms face difficulties in the learning process.

**CQL algorithm:** CQL is built on top of a Soft-Actor Critic algorithm (SAC) (Haarnoja et al., 2018), which employs soft-policy iteration of a stochastic policy (Haarnoja et al., 2017). A policy entropy regularization term is added to the policy improvement step in Eq. (2) as

$$\pi_{i+1} \leftarrow \arg\max_{\pi} \mathbb{E}_{s_t \sim \mathcal{D}}\left[\hat{Q}\big(s_t, \pi_i(a_t|s_t)\big) - \alpha \log \pi_i(a_t|s_t)\right] \quad . \tag{3}$$

where $\alpha$ either is a fixed hyperparameter or may be chosen to be trainable. CQL reduces the overestimation of state-values - in particular those out-of distribution from $\mathcal{D}$. It achieves this by regularizing the Q-function in Eq. (1) by a term minimizing its values over out of distribution randomly sampled action. In the following $a_{t+1}$ is given by the prediction of the policy $\pi_i(a_{t+1}|s_{t+1})$ and $\tilde{\alpha}$ is a hyperparameter balancing the regulizer term. The policy optimisation step is given by

$$\hat{Q}_{i+1} \leftarrow \arg\min_{Q} \left( \mathbb{E}_{s_t, a_t, s_{t+1} \sim \mathcal{D}}\left[\Big(r_t + \gamma \hat{Q}_i(s_{t+1}, a_{t+1}) - Q_i(s_t, a_t)\Big)^2\right] \right.$$
$$\left. + \tilde{\alpha} \mathbb{E}_{s_t \sim \mathcal{D}}\left[\log \sum_a \exp\big(Q_i(s_t, a)\big) - \mathbb{E}_{a \sim \pi(s_t)}\big[Q_i(s_t, a)\big]\right]\right) \tag{4}$$

## 2.2 KOOPMAN THEORY

Historically, the Koopman theoretic perspective of dynamical systems was introduced to describe the evolution of measurements of Hamiltonian systems (Koopman, 1931; Mezić, 2005). The underlying dynamic of most modern reinforcement learning tasks is of nonlinear nature, i.e. the agents actions lead to changes of it state described by a complex non-linear dynamical system. In contrast to linear systems which are completely characterized by their spectral decomposition non-linear systems lack such a unified characterisation. The Koopman operator theoretic framework describes nonlinear dynamics via a linear infinite-dimensional Koopman operator and thus inherits certain tools applicable to linear control systems (Mauroy et al., 2020; Kaiser et al., 2021). In practice one aims to find a finite-dimensional representation of the Koopman operator which is equivalent to obtaining a coordinate transformations in which the nonlinear dynamics are approximately linear. A general non-affine control system is governed by the system of non-linear ordinary differential equations (ODEs) as

$$\dot{s} = f(s, a) \tag{5}$$

where $s$, is the n-dimensional state vector, $a$ the m-dimensional action vector with $(s, a) \in \mathcal{S} \times \mathcal{A}$ the state-action-space. Moreover, $\dot{s} = \partial_t s$ is the time derivative, and $f$ is some general non-linear - at least $\mathcal{C}^1$-differentiable - vector valued function. For a discrete time system, Eq. (5) takes the form

$$s_{t+1} = F(s_t, a_t) \tag{6}$$

where $s_t$ denotes the state at time $t$ where F is at least $\mathcal{C}^1$-differentiable vector valued function.

**Definition 1 (Koopman operator)** *Let $\mathcal{K}(\mathcal{S} \times \mathcal{A})$ be the (Banach) space of all measurement functions (observables). Then the Koopman operator $\mathcal{K} : \mathcal{K}(\mathcal{S} \times \mathcal{A}) \rightarrow \mathcal{K}(\mathcal{S} \times \mathcal{A})$ is defined by*

$$\mathcal{K}g(s_t, a_t) = g\big(F(s_t, a_t), a_{t+1}\big) = g(s_{t+1}, a_{t+1}) , \quad \forall g \in \mathcal{K}(\mathcal{S} \times \mathcal{A}) \tag{7}$$

*where $g : \mathcal{S} \times \mathcal{A} \rightarrow \mathbb{R}$.*

Many systems can be modeled by a bilinearisation where the action enters the controlling equations 5 linearly as $f(s, a) = f_0(s) + \sum_i^m f_i(s)a_i$ for $f_i$, $i = 0, \ldots, m$ i.e. $\mathcal{C}^1$-differentiable-vector valued functions. In that case the action of the Koopman operator takes the simple form

$$g(s_{t+1}) = \mathcal{K}(a)g(s_t) = \left(\mathcal{K}_0 + \sum_i^m \mathcal{K}_i a_i\right)g(s_t) , \quad \forall g \in \mathcal{K}(\mathcal{S}) , \tag{8}$$

where $\mathcal{K}(\mathcal{S})$ is a (Banach) space of measurement functions $\mathcal{K}_0, \mathcal{K}_i$ decompose the Koopman operator in a the free and forcing-term, respectively. Details on the existence of a representation as in equation 8 are discussed in Goswami & Paley (2017). Associated with a Koopman operator is its eigenspectrum, that is, the eigenvalues $\lambda$, and the corresponding eigenfunctions $\varphi_\lambda(s, a)$, such that $[\mathcal{K}\varphi_\lambda](s, a) = \lambda\varphi_\lambda(s, a)$. In practice one derives a finite set of observable $\vec{g} = (g_1, \ldots, g_N)$ in which case the approximation to the Koopman operator admits a finite-dimensional matrix representation. The $N \times N$ matrix representing the Koopman operator may be diagonalized by a matrix $U$ containing the eigen-vectors of $\mathcal{K}$ as columns. In which case the eigen-functions are derived by $\vec{\varphi} = U\vec{g}$ and one infers from Eq. (5) that $\dot{\varphi}_i = \lambda_i\varphi_i$, for $i = 1, \ldots, N$ with eigenvalues $\lambda_{i=1,\ldots,N}$. These ODEs admit simple solutions for their time-evolution namely the exponential functions $\exp(\lambda_i t)$..

## 3 THE KOOPMAN FORWARD FRAMEWORK

We initiate our discussion with a focus on symmetries in dynamical control systems in Subsection 3.1 where we additionally present some theoretical results. We then proceed in Subsection 3.2 by presenting the Koopman forward framework for Q-learning based on CQL. Moreover, we discuss the algorithm as well as the Koopman Forward model's deep neural network architecture.

**Overview:** Our theoretical results culminate in Theorem 3.4 and 3.5, which provides a road-map on how specific symmetries of the dynamical control system are to be inferred from a VAE forward prediction model. In particular, Theorem 3.4 guarantees that the procedure leads to symmetries of the system (at least locally) and Theorem 3.5 that actual new data points can be derived by applying symmetry transformations to existing ones. Practically the VAE model parametrized by a neural net is trained data from one of many behavior policies and is thus learning an approximate dynamics. The theoretical limitations are twofold, firstly the theorems only hold for dynamical systems with differentiable state-transitions; secondly, we employ a Bilinearisation Ansatz for the Koopman operator of the system. Practically many RL environments incorporate dynamics with discontinuous "contact" events where the Bilinearisation Ansatz may not be applicable. However, empirically we find that our approach nevertheless is successful for environments with "contact" events. The latter does not affect the performance significantly (see Appendix D.1).

### 3.1 SYMMETRIES OF DYNAMICAL CONTROL SYSTEM

In general, a symmetry group $\Sigma$ of the state space may be any subgroup of the group of isometrics of the Euclidean space $\mathbb{E}^n$. We restrict oneself to an Euclidean state space, however in general one may consider Riemannian manifolds, see e.g. Giannakis (2019). In particular, in our work we also consider $\Sigma$ invariant compact subsets of Euclidean space. Moreover, relevant for a system of ODEs are local Lie symmetries as well as symmetries of ODEs.

**Definition 2 (Local Lie Group)** *A parametrized set of transformations $\sigma^\epsilon : \mathcal{S} \to \mathcal{S}$ with $s \mapsto \tilde{s}(s, \epsilon)$ for $\epsilon \in (\epsilon_{low}, \epsilon_{high})$ where $\epsilon_{low} < 0 < \epsilon_{high}$ is a one-parameter local Lie group if*

1. *$\sigma^0$ is the identity map i.e for $\epsilon = 0$ such that $\tilde{s}(s, 0) = s$.*

2. *$\sigma^{\epsilon_1}\sigma^{\epsilon_2} = \sigma^{\epsilon_2}\sigma^{\epsilon_1} = \sigma^{\epsilon_1+\epsilon_2}$ for every $|\epsilon_1|, |\epsilon_2| \ll 1$.*

3. *$\tilde{s}(s, \epsilon)$ admits a Taylor series expansion in $\epsilon$, i.e. in a neighbourhood of $s$ determined by $\epsilon = 0$ as $\tilde{s}(s, \epsilon) = s + \epsilon\,\zeta(s) + \mathcal{O}(\epsilon^2)$.*

The points (1) and (2) in definition 2 imply the existence of an inverse element $\sigma^{\epsilon^{-1}} = \sigma^{-\epsilon}$ for $|\epsilon| \ll 1$. Moreover, note that a system of ODEs may be represented by the set of its solutions.

**Definition 3 (Symmetries of ODEs)** *A symmetry of a system of ODEs on a locally-smooth structure (such as $\mathbb{E}^n$) is a locally-defined diffeomorphism that maps the set of all solutions to itself.*

**Definition 4 (Equivariant Dynamical System)** *Consider the dynamical system $\dot{s} = f(s)$ and let $\Sigma$ be a group acting on the state-space $\mathcal{S}$. Then the system is called $\Sigma$-equivariant if $f(\sigma \cdot s) = \sigma \cdot f(s)$, for $s \in \mathcal{S}$, $\forall \sigma \in \Sigma$. For a discrete time dynamical system $s_{t+1} = F(s_t)$ one defines equivariance analogously, namely if $F(\sigma \cdot s_t) = \sigma \cdot F(s_t)$, for $s_t \in \mathcal{S}$, $\forall \sigma \in \Sigma$.*

Note that as a local Lie group may satisfy the group axioms for sufficiently small parameters values it may not be a group on the entire set. And moreover not every diffeomorphism is also an isometry. Let us start by introducing symmetries in the simple context of dynamical system without control (Sinha et al., 2020) given by $\dot{s} = f(s)$ where we use analog notations as in Eq. (5). The Koopman operator of equivariant dynamical system is reviewed in the appendix A. Let us next turn to the case relevant for RL, namely control systems. In the remainder of this section we focus on dynamical systems given as in Eq. (6) and Eq. (8).

**Definition 5 (Action Equivariant Dynamical System)** *Let $\bar{\Sigma}$ be a group acting on the state-action-space $\mathcal{S} \times \mathcal{A}$ of a general control system as in Eq. (6) such that it acts as the identity operation on $\mathcal{A}$ i.e. $\sigma \cdot (s_t, a_t) = (\sigma|_{\mathcal{S}} \cdot s_t, a_t)$, $\forall \sigma \in \bar{\Sigma}$. Then the system is called $\bar{\Sigma}$-action-equivariant if*

$$F(\sigma \cdot (s_t, a_t)) = \sigma|_{\mathcal{S}} \cdot F(s_t, a_t), \quad \text{for } (s_t, a_t) \in \mathcal{S} \times \mathcal{A}, \quad \forall \sigma \in \bar{\Sigma}. \tag{9}$$

**Lemma 3.1** *The map $\bar{\divideontimes} : \bar{\Sigma} \times \mathcal{K}(\mathcal{S} \times \mathcal{A}) \to \mathcal{K}(\mathcal{S} \times \mathcal{A})$ given by $(\sigma \bar{\divideontimes} g)(s, a) \longmapsto g(\sigma^{-1} \cdot s, a)$ defines a group action on the Koopman space of observables $\mathcal{K}(\mathcal{S} \times \mathcal{A})$.*

**Theorem 3.2** *A Koopman operator $\mathcal{K}$ of a $\bar{\Sigma}$-action-equivariant system $s_{t+1} = F(s_t, a_t)$ satisfies*

$$[\sigma \bar{\divideontimes}(\mathcal{K}g)](s_t, a_t) = [\mathcal{K}(\sigma \bar{\divideontimes}g)](s_t, a_t). \tag{10}$$

In particular, it is easy to see that the biliniarisation in Eq. (8) is $\Sigma$-action-equivariant if $f_i(\sigma|_{\mathcal{S}} \cdot s) = \sigma|_{\mathcal{S}} \cdot f_i(s)$, $\forall i = 0, \ldots, m$. Let us thus turn our focus on the relevant case of a control system $s_{t+1} = \tilde{F}(s_t, a_t)$ which admits a Koopman operator description as

$$g(s_{t+1}) = \mathcal{K}(a_t)g(s_t), \quad \text{for } a_t \in \mathcal{A}, \forall g \in \mathcal{K}(\mathcal{S}), \tag{11}$$

where $\{\mathcal{K}(a)\}_{a \in \mathcal{A}}$ is a family of operators with analytical dependence on $a \in \mathcal{A}$. Note that the bilinearisation in Eq. (8) is a special case of Eq. (11). Furthermore, let $\{U(a)\}_{a \in \mathcal{A}}$ be a family of invertible operators s.t. $U(a) : \mathcal{K}(\mathcal{S}) \to \mathcal{F}(\mathcal{S} \times \mathcal{A})$ is a mapping to the (Banach) space of eigenfunctions $\mathcal{F}(\mathcal{S} \times \mathcal{A})$ with eigenfunctions $\varphi(s, a) := U(a)g(s)$ which obeys $U(a)\mathcal{K}(a)U(a)^{-1} = \Lambda(a)$, with $\Lambda(a)\varphi(s, a) = \lambda_\varphi(a)\varphi(s, a)$ and where $\lambda_\varphi(a) : \mathcal{A} \to \mathbb{R}$. The existence of such operators on an infinite dimensional space requires the Koopman operator in Eq. (11) to be self-adjoint or a finite-dimensional (approximate) matrix representation to be diagonalizable.[1]

**Lemma 3.3** *The map $\hat{\phi} : \bar{\Sigma} \times \mathcal{K}(\mathcal{S}) \times \{U(a)\}_{a \in \mathcal{A}} \to \mathcal{K}(\mathcal{S})$ given by*

$$(\sigma_a \hat{\divideontimes} g)(s) \longmapsto \left( U^{-1}(a)\left(\sigma \bar{\divideontimes}(U(a)g)\right)\right)(s) = g(\sigma^{-1} \cdot s), \tag{12}$$

*defines a group action on the Koopman space of observables $\mathcal{K}(\mathcal{S})$. Where $\bar{\divideontimes}$ is defined analog to Lemma 3.1 but acting on $\mathcal{F}(\mathcal{S} \times \mathcal{A})$ instead of $\mathcal{K}(\mathcal{S} \times \mathcal{A})$ by $(\sigma \bar{\divideontimes}\varphi)(s, a) \longmapsto \varphi(\sigma^{-1} \cdot s, a)$. We refer to a control system in Eq. (11) admitting a $\hat{\phi}$-symmetry as $\hat{\Sigma}$-action-equivariant.*

**Theorem 3.4** *Let $s_{t+1} = \tilde{F}(s_t, a_t)$ be a $\hat{\Sigma}$-action-equivariant control system with a symmetry action as in Lemma 3.3 which furthermore admits a Koopman operator representation as*

$$g(s_{t+1}) = \mathcal{K}(a_t)g(s_t), \quad \text{for } a_t \in \mathcal{A}, \forall g \in \mathcal{K}(\mathcal{S}). \tag{13}$$

*Then*

$$\left[\sigma_{a_t} \hat{\divideontimes}(\mathcal{K}(a_t)g)\right](s_t) = \left[\mathcal{K}(a_t)\left(\sigma_{a_t} \hat{\divideontimes}g\right)\right](s_t). \tag{14}$$

*Moreover, a control system obeying equations 13 and 14 is $\hat{\Sigma}$-action-equivariant locally if $g^{-1}$ exists for a neighborhood of $s_t$, i.e. then $\sigma \cdot \tilde{F}(s_t, a_t) = \tilde{F}(\sigma \cdot (s_t, a_t))$.*

---

[1] See Appendix A for the details. We found that, in practice the Koopman operator leaned by the neural nets is diagonalizable almost everywhere.

Let us next provide theoretical statements on how data-points may be shifted by symmetry transformations of solutions of ODEs. To establish an easier connection to the next section we introduce the notation Let $E : \mathcal{S} \to \mathcal{K}(S)$ and $D : \mathcal{K}(S) \to \mathcal{S}$ denote the $\mathcal{C}^1$-differentiable encoder and decoder to and from the finite-dimensional Koopman space approximation, respectively, i.e. $E \circ D = D \circ E = id$.

**Theorem 3.5** *Let $s_{t+1} = \tilde{F}(s_t, a_t)$ be a control system as in equation 13 and $\sigma_{a_t}$ an operator obeying equation 14. Then $\sigma_{a_t}^\epsilon : (s_t, s_{t+1}, a_t) \longmapsto (\tilde{s}_t, \tilde{s}_{t+1}, a_t)$ with*

$$\tilde{s}_t = D\Big( \big(\mathbb{1} + \epsilon \, \sigma_{a_t}\big) \hat{\ast} E(s_t)\Big) \,, \quad \tilde{s}_{t+1} = D\Big( \big(\mathbb{1} + \epsilon \, \sigma_{a_t}\big) \hat{\ast} E(s_{t+1})\Big) \tag{15}$$

*is a one-parameter local Lie symmetry group of ODEs. In other words one can use a symmetry transformation to shift both $s_t$ as well as $s_{t+1}$ such that $\tilde{s}_{t+1} = \tilde{F}(\tilde{s}_t, a_t)$.* [2]

### 3.2 KFC ALGORITHM

In the previous section we laid the theoretical foundation for the KFC-algorithm by providing a raod-map on how to derive symmetries of dynamical control systems based on a Koopman latent space representation. The goal is to generate new data points for the RL algorithm at training-time as Eq. (15) in Theorem 3.5. The reward $r_t$ is not part of the symmetry shift process and will just remain unchanged as an assertion.

**On the power of using symmetries:** Let us emphasize the practical advantage of employing symmetries to generate augmented data points. It is evident from Theorem 3.5 that a symmetry transformation shifts both $s_t$ as well as $s_{t+1}$ which evades the necessity of forecasting states. Thus the use of an inaccurate fore-cast model is avoided and the accuracy and generalisation capabilities of the VAE are fully utilized. The magnitude of the induced shift is controlled by the parameter $\epsilon \ll 1$ such that $|s - \tilde{s}| = \mathcal{O}(\epsilon)$ to limit out-of-distribution generalisation errors . Algorithmically the symmetry maps are derived in two distinct ways which we denote **KFC** and **KFC++**. The latter, constitute a simple starting point to extract symmetries from our setup.[3]

**Limitations:** Theorems 3.4 and 3.5 imply that if an operator commutes with the Koopman operator we can find a local symmetry group. However, global conditions may not be easily inferred. Moreover, in practice one has that $D \circ E \approx id$. Thus all assumptions of theorem 3.5 hold approximately.

**The Koopman forward model:** The KFC algorithm requires pre-training of a Koopman forward model $\mathcal{F} : \mathcal{S} \times \mathcal{A} \to \mathcal{S}$ which is closely related to a VAE architecture as

$$\mathcal{F}^c(s_t, a_t) = \begin{cases} D\big(E(s_t)\big) = s_t & \text{if } c = 0 \text{: VAE} \\ D\Big( \big(\mathcal{K}_0 + \sum_{i=1}^m \mathcal{K}_i \, a_{t,i}\big) E(s_t) \Big) = s_{t+1} & \text{if } c = 1 \text{: forward prediction} \end{cases} \tag{16}$$

where both of $E$ and $D$ are approximated by Multi-Layer-Perceptrons (MLP's) and the bilinear Koopman-space operator approximation are implemented by a single fully connected layer for $\mathcal{K}_i, \, i = 0, \dots, m$, respectively. The model is trained on batches of the offline data-set tuples $(s_{t+1}, s_t, a_t)$ and optimized via an additive loss-function of the VAE and the forward prediction part of the model. We refer the reader to Appendix C for details.

We integrate our framework into a specific Q-learning algorithm (CQL). In practice the Koopman latent space representation is N-dimensional (finite). The Koopman operator and the symmetry generators admit matrix representations; matrix multiplication replaces the mapping $\hat{\ast}$. Following in the footsteps of (Sinha et al., 2021) our approach leaves the policy improvement Eq. (3) unchanged but modifies the policy optimisation step Eq. (4) as

$$\hat{Q}_{i+1} \leftarrow \arg\min_Q \Bigg( \mathop{\mathbb{E}}_{s_t, a_t, s_{t+1} \sim \mathcal{D}} \bigg[ \Big( r_t + \gamma \hat{Q}_i\big(\sigma_{a_t}^\epsilon(\tilde{s}_{t+1}|s_{t+1}), a_{t+1}\big) - Q_i\big(\sigma_{a_t}^\epsilon(\tilde{s}_t|s_t), a_t\big) \Big)^2 \bigg]$$
$$+ \tilde{\alpha} \mathop{\mathbb{E}}_{s_t \sim \mathcal{D}} \Big[ \log \sum_a \exp\big(Q_i(s_t, a)\big) - \mathop{\mathbb{E}}_{a \sim \pi(s_t)}\big[Q_i(s_t, a)\big] \Big] \Bigg) \tag{17}$$

---

[2]No assumptions on the Koopman operator are imposed. Moreover, note that the equivalent theorem holds when $\epsilon \, \sigma_{a_t} \to \sum_{I=1} \epsilon_I \, \sigma_{a_t}^I$ to be a local N-parameter Lie group.

[3]More elaborate studies employing the extended literate on Koopman spectral analysis are desirable. See appendix B for details on numerical errors.

The state-space symmetry generating function $\sigma_a^\epsilon : \mathcal{S} \to \mathcal{S}$ depends on the normally distributed random variables $\epsilon$. Note that we only modify the Bellman error of Eq. (17) and leave the CQL specific regulizer untouched. We study two distinct cases which differ on an algorithmic level[4]

$$\textbf{KFC} \;\; \sigma_{\mathbf{a_t}}^\epsilon : s \mapsto \tilde{s} = D\Big(\big(\mathbb{1} + \epsilon\sigma_{a_t}\big)E(s)\Big) \qquad \textbf{KFC++} \;\; \sigma_{\mathbf{a_t}}^\epsilon : s \mapsto \tilde{s} = D\Big(\big(\mathbb{1} + \sigma_{a_t}(\vec{\epsilon})\big)E(s)\Big),$$

$$\text{where} \;\; \sigma_{a_t}(\vec{\epsilon}) = \mathfrak{Re}\big(U(a_t)\operatorname{diag}(\epsilon_1,\ldots,\epsilon_N)\,U^{-1}(a_t)\big), \tag{18}$$

and $U(a_t)$ diagonalizes the Koopman operator with eigenvalues $\lambda_i$, $i = 1,\ldots,N$ i.e. the latter can be expressed as $\mathcal{K}(a_t) = U(a_t)\operatorname{diag}(\lambda_1,\ldots,\lambda_N)U^{-1}(a_t)$. Note that we abuse our notation as $E(s) \equiv \vec{g}(s) = [g_1(s),\ldots,g_N(s)]$ i.e. the encoder provides the Koopman space observables. From theorem 3.5 one infers that in order for Eq. (18) to be a local Lie symmetry group of the approximate ODEs captured by the neural net Eq. (16) one needs $\sigma_{a_t}$ to commute with the approximate Koopman operator in Eq. (16).

For the KFC option in Eq. (18) the symmetry generator matrix $\sigma_a$ is obtained by solving the equation $\sigma_{a_t} \cdot \mathcal{K}(a_t) - \mathcal{K}(a_t) \cdot \sigma_{a_t} = 0$ which may be accomplished by employing a Sylvester algorithm (syl). For KFC++ we compute the eigen-vectors of $\mathcal{K}(a)$ which then constitute the columns of $U(a)$. Thus in particular one infers that $[\sigma_{a_t}(\vec{\epsilon}), \mathcal{K}(a_t)] = 0$. The latter, is solved by construction of $\sigma_a(\vec{\epsilon})$ which commutes with the Koopman operator for all values of the random variables.[5] The advantage of KFC is that it is computationally less expensive than KFC++ , however it provides less freedom to explore different symmetry directions than the latter. We employ our symmetry shift only with probability $p_K$; otherwise we use a random normally distributed state shift.

## 4 Empirical evaluation

In this section, we will first experiment with the popular D4RL benchmark commonly used for offline RL (Fu et al., 2021). The benchmark covers various different tasks such as locomotion tasks with Mujoco Gym (Brockman et al., 2016), tasks that require hierarchical planning such as antmaze, and other robotics tasks such as kitchen and adroit (Rajeswaran et al., 2017). Furthermore, similar to S4RL (Sinha et al., 2021), we perform experiments on 6 different challenging robotics tasks from MetaWorld (Yu et al., 2019) and RoboSuite (Zhu et al., 2020). We compare KFC to the baseline CQL algorithm (Kumar et al., 2020), and two best performing augmentation variants from S4RL, S4RL-$\mathcal{N}$ and S4RL-adv (Sinha et al., 2021). We use the exact same hyperparameters as proposed in the respective papers. Furthermore, similar to S4RL, we build KFC on top of CQL (Kumar et al., 2020) to ensure conservative Q-estimates during for policy evaluation.

### 4.1 D4RL benchmarks

We present results in the benchmark D4RL test suite and report the normalized return in Table 1. We see that both KFC and KFC++ consistently outperform both the baseline CQL and S4RL across multiple tasks and data distributions. Outperforming S4RL-$\mathcal{N}$ and S4RL-adv on various different types of environments suggests that KFC and KFC++ fundamentally improves the data augmentation strategies discussed in S4RL. KFC-variants also improve the performance of learned agents on challenging environments such as antmaze: which requires hierarchical planning, kitchen and adroit tasks: which are sparse reward and have large action spaces. Similarly, KFC-variants also perform well on difficult data distributions such as "medium-replay": which is a collected by simply using all the data that the policy encountered while training base SAC policy, and "-human" which is collected using human demonstrations on robotic tasks which results in a non-Markovian behaviour policy (more details can be found in the D4RL manuscript (Fu et al., 2021)). Furthermore, to our knowledge, the results for KFC++ are state-of-the-art in policy learning from D4RL datasets for most environments and data distributions.

We do note that S4RL outperforms KFC on the "-random" split of the data distributions, which is expected as KFC depends on learning a simple dynamics model of the data to use to guide the data augmentation strategy. Since the "-random" split consists of random actions in the environment, our simple model is unable to learn a useful dynamics model. For ablation studies see appendix D.

---

[4]Eq. (18) holds for both $s_t$ as well as $s_{t+1}$ thus we have deliberately dropped the subscript. Moreover, note that in case (II) the symmetry transformations are of the form $\sigma_{a_t}(\vec{\epsilon}) = \sum_{I=1} \epsilon_I \, \sigma_{a_t}^I$.

[5]$[\cdot, \cdot]$ denotes the commutator of two matrices. The Koopman operators eigenvalues and eignevectors generically are $\mathbb{C}$-valued. However, the definition in Eq. (18) ensures that $\sigma_a(\vec{\epsilon})$ is a $\mathbb{R}$-valued matrix.

**Table 1:** We experiment with the full set of the D4RL tasks and report the mean normalized episodic returns over 5 random seeds using the same protocol as Fu et al. (2021). We compare against 3 competitive baselines including CQL and the two best performing S4RL-data augmentation strategies. We see that KFC and KFC++ consistently outperforms the baselines. We use the baseline numbers reported in Sinha et al. (2021).

| Domain | Task Name | CQL | S4RL-($\mathcal{N}$) | S4RL-(Adv) | KFC | KFC++ |
|---|---|---|---|---|---|---|
| AntMaze | antmaze-umaze | 74.0 | 91.3 | 94.1 | **96.9** | **99.8** |
| | antmaze-umaze-diverse | 84.0 | 87.8 | 88.0 | **91.2** | **91.1** |
| | antmaze-medium-play | 61.2 | 61.9 | 61.6 | 60.0 | **63.1** |
| | antmaze-medium-diverse | 53.7 | 78.1 | 82.3 | 87.1 | **90.5** |
| | antmaze-large-play | 15.8 | 24.4 | **25.1** | 24.8 | **25.6** |
| | antmaze-large-diverse | 14.9 | 27.0 | 26.2 | **33.1** | **34.0** |
| Gym | cheetah-random | 35.4 | **52.3** | **53.9** | 48.6 | 49.2 |
| | cheetah-medium | 44.4 | 48.8 | 48.6 | 55.9 | **59.1** |
| | cheetah-medium-replay | 42.0 | 51.4 | 51.7 | **58.1** | **58.9** |
| | cheetah-medium-expert | 62.4 | **79.0** | 78.1 | **79.9** | 79.8 |
| | hopper-random | **10.8** | **10.8** | 10.7 | 10.4 | **10.7** |
| | hopper-medium | 58.0 | 78.9 | 81.3 | 90.6 | **94.2** |
| | hopper-medium-replay | 29.5 | 35.4 | 36.8 | **48.6** | **49.0** |
| | hopper-medium-expert | 111.0 | 113.5 | 117.9 | 121.0 | **125.5** |
| | walker-random | 7.0 | **24.9** | **25.1** | 19.1 | 17.6 |
| | walker-medium | 79.2 | 93.6 | 93.1 | 102.1 | **108.0** |
| | walker-medium-replay | 21.1 | 30.3 | 35.0 | **48.0** | 46.1 |
| | walker-medium-expert | 98.7 | 112.2 | 107.1 | 114.0 | **115.3** |
| Adroit | pen-human | 37.5 | 44.4 | 51.2 | **61.3** | 60.0 |
| | pen-cloned | 39.2 | 57.1 | 58.2 | **71.3** | 68.4 |
| | hammer-human | 4.4 | 5.9 | 6.3 | 7.0 | **9.4** |
| | hammer-cloned | 2.1 | 2.7 | 2.9 | 3.0 | **4.2** |
| | door-human | 9.9 | 27.0 | 35.3 | 44.1 | **46.1** |
| | door-cloned | 0.4 | 2.1 | 0.8 | 3.6 | **5.6** |
| | relocate-human | **0.2** | **0.2** | **0.2** | **0.2** | **0.2** |
| | relocate-cloned | **-0.1** | **-0.1** | -0.1 | -0.1 | -0.1 |
| Franka | kitchen-complete | 43.8 | 77.1 | 88.1 | **94.1** | **94.9** |
| | kitchen-partial | 49.8 | 74.8 | 83.6 | 92.3 | **95.9** |

**(a)** MetaWorld Environments     **(b)** RoboSuite Environments

**Figure 2:** Results on challenging dexterous robotics environments using data collected using a similar strategy as S4RL (Sinha et al., 2021). We report the % of goals that the agent is able to reach during evaluation, where the goal is set by the environments. **We see that KFC and KFC++ consistently outperforms both CQL and the two best performing S4RL variants.**

## 4.2 METAWORLD AND ROBOSUITE BENCHMARKS

To further test the ability of KFC, we perform additional experiments on challenging robotic tasks. Following (Sinha et al., 2021), we perform additional experiments with 4 MetaWorld environments (Yu et al., 2019) and 2 RoboSuite environments (Zhu et al., 2020). We followed the same method to

collect the data as described in Appendix F of S4RL (Sinha et al., 2021), and report the mean percent of goals reached, where the condition of reaching the goal is defined by the environment.

We report the results in Figure 2, where we see that using KFC to guide the data augmentation strategy for a base CQL agent, we are able to learn an agent that performs significantly better. Furthermore, we see that for more challenging tasks such as "push" and "door-close" in the MetaWorld, KFC++ outperforms the base CQL algorithm and the S4RL agent by a significant margin. These set of experiments further highlight the ability of KFC to guide the data augmentation strategy.

## 5 RELATED WORKS

The use of data augmentation techniques in Q-learning has been discussed recently (Laskin et al., 2020b;a; Sinha et al., 2021). In particular, our work shares strong parallels with (Sinha et al., 2021). Our modification of the policy evaluation step of the CQL algorithm (Kumar et al., 2020) is analogous to the one in Sinha et al. (2021). However, the latter randomly augments the data while our augmentation framework is based on symmetry state shifts. Regarding the connection to world models (Ha & Schmidhuber, 2018). Here a VAE is used to decode the state information while a recurrent separate neural network predicts future states. Their latent representation is not of Koopman type. Also no symmetries and data-augmentations are derived.

Algebraic symmetries of the state-action space in Markov Decision Processes (MDP) originate (Balaraman & Andrew, 2004) an were discussed recently in the context of RL in (van der Pol et al., 2020). Their goal is to preserve the essential algebraic homomorphism symmetry structure of the original MDP while finding a more compact representation. The symmetry maps considered in our work are more general and are utilized in a different way. Symmetry-based representation learning (Higgins et al., 2018) refers to the study of symmetries of the environment manifested in the latent representation. The symmetries in our case are derived form the Koopman operator not the latent representation directly. In (Caselles-Dupré et al., 2019) the authors discuss representation learning of symmetries (Higgins et al., 2018) allowing for interactions with the environment. A Forward-VAE model which is similar to our Koopman-Forward VAE model is employed. However, our approach is based on theoretical results providing a road-map to derive explicit symmetries of the dynamical systems as well as their utilisation for state-shifts.

In (Sinha et al., 2020) the authors extend the Koopman operator from a local to a global description using symmetries of the dynamics. They do not discuss action-equivariant dynamical control systems nor data augmentation. In (Salova et al., 2019) the imprint of known symmetries on the block-diagonal Koopman space representation for non-control dynamical systems is discussed. This is close to the spirit of disentanglement (Higgins et al., 2018). Our results are on control setups and deriving symmetries. On another front, the application of Koopman theory in control or reinforcement learning has also been discussed recently. For example, Li et al. (2020) propose to use compositional Koopman operators using graph neural networks to learn dynamics that can quickly adapt to new environments of unknown physical parameters and produce control signals to achieve a specified goal. Kaiser et al. (2021) discuss the use of Koopman eigenfunction as a transformation of the state into a globally linear space where the classical control techniques is applicable. To the best of our knowledge, this paper is the first to discuss Koopman latent space for data augmentation.

## 6 CONCLUSIONS

In this work we proposed a symmetry-based data augmentation technique derived from a Koopman latent space representation. It enables a meaningful extension of offline RL datasets describing dynamical systems, i.e. further "exploration" without additional environment interactions. The approach is based on our theoretical results on symmetries of dynamical control systems and symmetry shifts of data. Both hold for systems with differentiable state transitions and with a Bilinearisation Ansatz for the Koopman operator. However, the empirical results show that the framework is successfully applicable beyond those limitations. We empirically evaluated our method on several benchmark offline reinforcement learning tasks D4RL, Metaworld and Robosuite and find that by using our framework we consistently improve the state-of-the-art of Q-learning algorithms.

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

## A    KOOPMAN THEORY AND SYMMETRIES

This section provides some additional introductory material on Koopman theory and symmetries of dynamical systems.

**Lemma A.1** *The map* $* : \Sigma \times \mathcal{K}(\mathcal{S}) \to \mathcal{K}(\mathcal{S})$ *given by* $(\sigma * g)(s) \longmapsto g(\sigma^{-1} \cdot s)$ *defines a group action on the Koopman space of observables* $\mathcal{K}(\mathcal{S})$.

**Theorem A.2** *Let* $\mathcal{K}$ *be the Koopman operator associated with a* $\Sigma$-*equivariant system* $s_{t+1} = F(s_t)$. *Then*
$$[\sigma * (\mathcal{K}g)](s) = [\mathcal{K}(\sigma * g)](s) \ . \tag{19}$$

Theorem A.2 states that for a $\Sigma$-equivariant system any symmetry transformation commutes with the Koopman operator. For the proof see (Sinha et al., 2020).

To proceed let us re-evaluate some information from the main text of this work. Let $\{U(a)\}_{a \in \mathcal{A}}$ be a family of invertible operators s.t. $U(a) : \mathcal{K}(\mathcal{S}) \to \mathcal{F}(\mathcal{S} \times \mathcal{A})$ is a mapping to the (Banach) space of eigenfunctions $\varphi(s, a) := U(a)g(s) \in \mathcal{F}(\mathcal{S} \times \mathcal{A})$. Which moreover obeys $U(a)\mathcal{K}(a)U(a)^{-1} = \Lambda(a)$, with $\Lambda(a)\varphi(s, a) = \lambda_\varphi(a)\varphi(s, a)$ and where $\lambda_\varphi(a) : \mathcal{A} \to \mathbb{R}$. The existence of such operators puts further restriction on the Koopman operator in Eq. (11). However, as our algorithm employs the finite-dimensional approximation of the Koopman operator i.e. its matrix representation this amounts simple for $\mathcal{K}(a)$ to be diagonalizable and $U(a)$ is the matrix containing its eigen-vectors as columns. To evolve a better understanding on the required criteria for the infinite-dimensional case we employ an alternative formulation of the so called spectral theorem below.

**Theorem A.3 (Spectral Theorem)** *Let* $\mathcal{K}$ *be a bounded self-adjoint operator on a Hilbert space* $\mathcal{H}$. *Then there is a measure space* $(\mathcal{S}, \Sigma, \mu)$ *and a real-valued essentially bounded measurable function* $\lambda$ *on* $\mathcal{S}$ *and a unitary operator* $U : \mathcal{H} \to L^2_\mu(\mathcal{S})$ *i.e.* $U^*U = U U^* = id$ *such that*
$$U \Lambda U^* = \mathcal{K} \ , \quad with \ \ [\Lambda\varphi](s) = \lambda(s)\varphi(s) \ . \tag{20}$$

In other words every bounded self-adjoint operator is unitarily equivalent to a multiplication operator. In contrast to the finite-dimensional case we need to slightly alter our criteria to $U(a)\mathcal{K}(a)U(a)^{-1} = \Lambda(a)$, with $\Lambda(a)\varphi(s, a) = \lambda_\varphi(a, s)\varphi(s, a)$ and where $\lambda_\varphi(s, a) : \mathcal{S} \times \mathcal{A} \to \mathbb{R}$. Concludingly, a sufficient condition for our criteria to hold in terms of operators on Hilbert spaces is that the Koopman operator $\mathcal{K}(a)$ is self-adjoint i.e. that
$$\mathcal{K}(a) = \mathcal{K}(a)^* \ . \tag{21}$$

## B    PROOFS

In this section we provide the proofs of the theoretical results of Section 3.

**Proof of Lemma A.1 and Theorem A.2**    The proofs of the lemma as well as the theorem can be found in (Sinha et al., 2020).

**Proof of Lemma 3.1:**    We aim to show that map $\bar{*} : \bar{\Sigma} \times \mathcal{K}(\mathcal{S} \times \mathcal{A}) \to \mathcal{K}(\mathcal{S} \times \mathcal{A})$ given by $(\sigma \bar{*} g)(s, a) \longmapsto g(\sigma^{-1} \cdot s, a)$ defines a group action on the Koopman space of observables $\mathcal{K}(\mathcal{S} \times \mathcal{A})$, where $\bar{\Sigma}$ defines the symmetry group of definition 5. Firstly, let $g \in \mathcal{K}(\mathcal{S} \times \mathcal{A})$ and $\sigma_0 \in \bar{\Sigma}$ be the identity element, then we see that it provides existence of an identity element of $\bar{*}$ by
$$(\sigma_0 \bar{*} g)(s, a) = g(\sigma_0^{-1} \cdot s, a) = g(s, a) \ .$$
Secondly, let $\sigma_1, \sigma_2 \in \bar{\Sigma}$ and $\odot$ denoting the group operation i.e. $\sigma_1 \odot \sigma_2 = \sigma_3 \in \bar{\Sigma}$. Then
$$\left(\sigma_2 \bar{*} (\sigma_1 \bar{*} g)\right)(s, a) \ \ = \sigma_2 \bar{*} \left(g(\sigma_1^{-1} \cdot s, a)\right) = g(\sigma_2^{-1} \cdot (\sigma_1^{-1} \cdot s), a) = g((\sigma_2^{-1} \odot \sigma_1^{-1}) \cdot s), a)$$

$$\overset{(I)}{=} g((\sigma_1 \odot \sigma_2)^{-1} \cdot s, a) = \left((\sigma_1 \odot \sigma_2) \bar{*} g\right)(s, a) = \left(\sigma_3 \bar{*} g\right)(s, a) \ ,$$

where in $(I)$ we have used the invertibility of the group property of $\bar{\Sigma}$. Lastly it follows analogously that for $\sigma, \sigma^{-1} \in \bar{\Sigma}$ that
$$\left(\sigma \bar{*} (\sigma^{-1} \bar{*} g)\right)(s, a) = \left((\sigma^{-1} \odot \sigma) \bar{*} g\right)(s, a) = \left(\sigma_0 \bar{*} g\right)(s, a) = g(s, a) \ .$$
Thus the existence of an inverse is established which concludes to show the group property of $\bar{*}$.

**Proof of Theorem 3.2:** We aim to show that with $\mathcal{K}$ being the Koopman operator associated with a $\bar{\Sigma}$-action-equivariant system $s_{t+1} = F(s_t, a_t)$. Then

$$\left[ \sigma\bar{\ast}(\mathcal{K}g) \right](s,a) = \left[ \mathcal{K}(\sigma\bar{\ast}g) \right](s,a) \ .$$

First of all note that by the definition of the Koopman operator of non-affine control systems it obeys

$$[\mathcal{K}g](s_t, a_t) = g(F(s_t, a_t), a_{t+1}) \ .$$

Using the latter one thus infers that

$$\left[ \sigma\bar{\ast}(\mathcal{K}g) \right](s_t, a_t) = \sigma\bar{\ast}g(F(s_t,a_t), a_{t+1}) = g(\sigma^{-1} \cdot F(s_t,a_t), a_{t+1}) \overset{(I)}{=} g(F(\sigma^{-1} \cdot s_t, a_t), a_{t+1}) \ ,$$

where in (I) we have used that it is a $\bar{\Sigma}$-action-equivariant system. Moreover, one finds that

$$g(F(\sigma^{-1} \cdot s_t, a_t), a_{t+1}) = \mathcal{K}g(\sigma^{-1} \cdot s_t, a_t) = \left[ \mathcal{K}(\sigma\bar{\ast}g) \right](s_t, a_t) \ ,$$

which concludes the proof.

**Proof of Lemma 3.3:** We aim to show that the map $\hat{\phi} : \bar{\Sigma} \times \mathcal{K}(\mathcal{S}) \times \{U(a)\}_{a \in \mathcal{A}} \to \mathcal{K}(\mathcal{S})$ given by

$$(\sigma_a \hat{\ast} g)(s) \longmapsto \left( U^{-1}(a)\left(\sigma\bar{\ast}(U(a)g)\right) \right)(s) = g(\sigma^{-1} \cdot s) \ ,$$

defines a group action on the Koopman space of observables $\mathcal{K}(\mathcal{S})$. Where $\bar{\ast}$ is defined analog to Lemma 3.1 but acting on $\mathcal{F}(\mathcal{S} \times \mathcal{A})$ instead of $\mathcal{K}(\mathcal{S} \times \mathcal{A})$ by $(\sigma\bar{\ast}\varphi)(s,a) \longmapsto \varphi(\sigma^{-1} \cdot s, a)$. First of all note that

$$\left( U^{-1}(a)\left(\bar{\sigma} \ast (U(a)g)\right) \right)(s) = U^{-1}(a)\left(\bar{\sigma} \ast \varphi(s,a)\right) = U^{-1}(a)\varphi(\sigma^{-1} \cdot s, a) = g(\sigma^{-1} \cdot s) \quad (22)$$

We proceed analogously as in the proof of Lemma 3.1 above. Firstly, let $g \in \mathcal{K}(\mathcal{S})$ and $\sigma_0 \in \bar{\Sigma}$ be the identity element then on infers from Eq. (22) that it provides existence of an identity element of $\hat{\phi}$ by

$$(\sigma_{a,0}\hat{\ast}g)(s) = g(\sigma_0^{-1} \cdot s) = g(s) \ ,$$

where we have used the notation

$$(\sigma_{a,i}\hat{\ast}g)(s) = \left( U^{-1}(a)\left(\sigma_i\bar{\ast}(U(a)g)\right) \right)(s) \ , \quad \text{for} \ \ i = 0, \dots \ .$$

Secondly, let $\sigma_1, \sigma_2 \in \bar{\Sigma}$ and $\odot$ denoting the group operation i.e. $\sigma_1 \odot \sigma_2 = \sigma_3 \in \bar{\Sigma}$. Then

$$\left( \sigma_{a,2}\hat{\ast}\left(\sigma_{a,1}\hat{\ast}g\right) \right)(s) \ = \sigma_{a,2}\hat{\ast}\left( g(\sigma_1^{-1} \cdot s) \right) = g(\sigma_2^{-1} \cdot (\sigma_1^{-1} \cdot s)) = g((\sigma_2^{-1} \odot \sigma_1^{-1}) \cdot s)$$

$$\overset{(I)}{=} g((\sigma_1 \odot \sigma_2)^{-1} \cdot s) = g(\sigma_3^{-1} \cdot s) = \left( \sigma_{a,3}\bar{\ast}g \right)(s) \ ,$$

where in $(I)$ we have used the invertibility of the group property of $\Sigma$. Lastly, it follows analogously that for $\sigma, \sigma^{-1} \in \bar{\Sigma}$ that

$$\left( \sigma\hat{\ast}\left(\sigma^{-1}\hat{\ast}g\right) \right)(s) = \left( (\sigma^{-1} \odot \sigma)\hat{\ast}g \right)(s) = \left( \sigma_0\hat{\ast}g \right)(s) = g(s) \ .$$

Thus the existence of an inverse is established which concludes to show the group property of $\hat{\phi}$.

**Proof of Theorem 3.4:** We aim to show twofold.

$\Longrightarrow$: Firstly, that with $s_{t+1} = \tilde{F}(s_t, a_t)$ be a $\hat{\Sigma}$-action-equivariant control system with a symmetry action as in Lemma 3.3 which furthermore admits a Koopman operator representation as

$$g(s_{t+1}) = \mathcal{K}(a_t)g(s_t) \ , \quad \text{for} \ a_t \in \mathcal{A}, \ \forall g \in \mathcal{K}(\mathcal{S}) \ .$$

Then

$$\left[ \sigma_{a_t}\hat{\ast}(\mathcal{K}(a_t)g) \right](s_t) = \left[ \mathcal{K}(a_t)\left(\sigma_{a_t}\hat{\ast}g\right) \right](s_t) \ .$$

**⟸:** Secondly, the converse. Namely, if a control system $s_{t+1} = \tilde{F}(s_t, a_t)$ obeys Eqs. (13) and (14), then it is $\hat{\Sigma}$-action-equivariant, i.e. $\sigma \cdot \tilde{F}(s_t, a_t) = \tilde{F}(\sigma \cdot (s_t, a_t))$. For notational simplicity we drop the subscripts referring to time in the remainder of this proof i.e. $s_t \to s$ and $a_t \to a$.

Let us start with the first implication i.e. $\Longrightarrow$.

First of all note that by the definition of the Koopman operator one has

$$\big[\mathcal{K}(a)g\big](s) = g\big(\tilde{F}(s, a)\big) \ .$$

Using the latter one infers that

$$
\begin{aligned}
\Big[\sigma_a \hat{*}(\mathcal{K}(a)g)\Big](s) &= \sigma_a \hat{*} g(\tilde{F}(s, a)) \ , \\
&= U^{-1}(a)\Big(\sigma \bar{*}(U(a)g(\tilde{F}(s, a)))\Big) \ , \\
&= U^{-1}(a)\Big(\sigma \bar{*}\varphi(\tilde{F}(s, a), a)\Big) \ , \\
&= U^{-1}(a)\Big(\varphi(\sigma^{-1} \cdot \tilde{F}(s, a), a))\Big) \ , \\
&= U^{-1}(a)\Big(\varphi(\tilde{F}(\sigma^{-1} \cdot s, a), a))\Big) \ , \\
&= g\big(\tilde{F}(\sigma^{-1}s, a)\big) \ .
\end{aligned}
$$

Moreover, one derives that

$$
\begin{aligned}
g\big(\tilde{F}(\sigma^{-1}s, a)\big) &= \mathcal{K}(a)g(\sigma^{-1}s) \ , \\
&= \mathcal{K}(a) \underbrace{U^{-1}(a)\,U(a)}_{=id} g(\sigma^{-1}s) \ , \\
&= \mathcal{K}(a)\,U^{-1}(a)\,\varphi(\sigma^{-1}s, a) \ , \\
&= \mathcal{K}(a)\Big(U^{-1}(a)\,(\sigma \bar{*}\varphi(s, a))\Big) \ , \\
&= \mathcal{K}(a)\Big(U^{-1}(a)\,(\sigma \bar{*}(U(a)g(s)))\Big) \ , \\
&= \Big[\mathcal{K}(a)\,(\sigma_a \hat{*}g)\Big](s) \ ,
\end{aligned}
$$

which concludes the proof of the first part of the theorem. Let us next show the converse implication i.e. $\Longleftarrow$. For this case it is practical to use the discrete time-system notation explicitly. Let $\sigma \in \Sigma$ be a symmetry of the state-space and be $\tilde{s}_t = \sigma \cdot s_t$ and $\tilde{s}_{t+1} = \sigma \cdot s_{t+1}$ the $\sigma$-shifted states. Then

$$
\begin{aligned}
g\big(s_{t+1}\big) &= g\big(\tilde{F}(s_t, a_t)\big) \ , \\
&= \mathcal{K}(a_t)\,g(s_t) \ , \\
&= \mathcal{K}(a_t)\,g\big(\sigma^{-1} \cdot \tilde{s}_t\big) \ , \\
&= \mathcal{K}(a_t)\Big(\sigma_a \hat{*}g\big(\tilde{s}_t\big)\Big) \ , \\
&= \Big[\mathcal{K}(a_t)(\sigma_a \hat{*}g)\Big](\tilde{s}_t) \ , \\
&\overset{(I)}{=} \Big[\sigma_a \hat{*}(\mathcal{K}(a_t)g)\Big](\tilde{s}_t) \ .
\end{aligned}
$$

Thus in particular

$$\sigma_a^{-1}\hat{*}g\big(s_{t+1}\big) = \Big[\sigma_a^{-1}\hat{*}\big(\sigma_a \hat{*}(\mathcal{K}(a_t)g)\big)\Big](\tilde{s}_t) \overset{Lemma\ 3.3}{=} \Big[\mathcal{K}(a_t)g\Big](\tilde{s}_t) \ ,$$

where in (I) we have used that the symmetry operator commutes with the Koopman operator. Moreover, one finds that

$$\sigma_a^{-1} \hat{*} g\big(s_{t+1}\big) = g\Big(\big(\sigma^{-1}\big)^{-1} \cdot s_{t+1}\Big) = g\big(\sigma \cdot s_{t+1}\big) = g\big(\tilde{s}_{t+1}\big) \ ,$$

from which one concludes that

$$g(\tilde{s}_{t+1}) = \big[\mathcal{K}(a_t)g\big](\tilde{s}_t) = g\big(\tilde{F}(\tilde{s}_t, a_t)\big) \ .$$

Finally, we use the invertibility of the Koopman space observables i.e. $g^{-1}(g(s)) = s$ to infer

$$\tilde{s}_{t+1} = \tilde{F}(\tilde{s}_t, a_t) \ .$$

However, in general s the Koopman space observables are not invertible globally. The "inverse function theorem" guarantees the existence of a local inverse if $g(s)$ is $\mathcal{C}^1$ differentiable for maps between manifolds of equal dimensions. However, we assume inevitability locally.

$$\tilde{s}_{t+1} = \sigma \cdot s_{t+1} \ \Rightarrow \ \tilde{F}(\tilde{s}_t, a_t) = \sigma \cdot \tilde{F}(s_t, a_t) \ \Rightarrow \ \tilde{F}(\sigma \cdot s_t, a_t) = \sigma \cdot \tilde{F}(s_t, a_t) \ , \quad (23)$$

which at last concludes our proof of the second part of the theorem.

**Extension of Theorem 3.5:**  Moreover, one may account for practical limitations i.e. an error by the assumption $[\sigma_{a_t}^\epsilon, \mathcal{K}(a_t)] = \epsilon_a \mathbb{1}$. One then finds that $\tilde{s}_{t+1} = \tilde{F}(\tilde{s}_t, a_t) + \mathcal{O}(\epsilon_a)$. Thus the error becomes suppressed when $\epsilon_a \ll \epsilon$. The error may be due to practical limitations of capturing the true dynamics as well as the symmetry map.

**Proof of Theorem 3.5:**  We will show here theorem 3.5 as well as the extension mentioned in the paragraph prior to this proof. Let $s_{t+1} = \tilde{F}(s_t, a_t)$ be a $\hat{\Sigma}$-action-equivariant control system as in Theorem 3.4. For symmetry maps

$$\tilde{s}_t = D\Big( \big(\mathbb{1} + \epsilon \sigma_{a_t}\big) \hat{*} E(s_t)\Big) \ , \ \ \tilde{s}_{t+1} = D\Big( \big(\mathbb{1} + \epsilon \sigma_{a_t}\big) \hat{*} E(s_{t+1})\Big) \ . \quad (24)$$

we aim to show that one can use a symmetry transformation to shift both $s_t$ as well as $s_{t+1}$

$$\sigma_{a_t}^\epsilon : \ \big(s_t, s_{t+1}, a_t\big) \ \longmapsto \ \big(\tilde{s}_t, \tilde{s}_{t+1}, a_t\big) \ , \ \ \text{s.t.} \ \ \tilde{s}_{t+1} = \tilde{F}(\tilde{s}_t, a_t) \ . \quad (25)$$

From equation 24 and the definition of the Koopman operator one infers that[6]

$$\tilde{s}_{t+1} = D\Big( \sigma_{a_t} \hat{*} E(s_{t+1})\Big) = D\Big( \sigma_{a_t} \hat{*} \mathcal{K}(a_t) E(s_t)\Big) \ . \quad (26)$$

By using that $[\sigma_{a_t}, \mathcal{K}(a_t)] = 0$ and equation 24 one finds that

$$\tilde{s}_{t+1} = D\Big( \mathcal{K}(a_t)\sigma_{a_t} \hat{*} E(s_t)\Big) = D\Big( \mathcal{K}(a_t) \underbrace{E(D(\sigma_{a_t} \hat{*} E(s_t)))}_{=\mathbb{1}}\Big) = D\Big( \mathcal{K}(a_t)E(\tilde{s}_t)\Big) \ , \quad (27)$$

which concludes the proof of the first part of the theorem with $\sigma_{a_t} \to (\mathbb{1} + \epsilon \sigma_{a_t})$.

**Local diffeomorphism.**  Per definition the maps $D, E$ are differentiable and invertible which implies that they provide a local diffeomorphism from and to Koopman-space. Also the linear map i.e. a matrix multiplication by $(\mathbb{1} + \epsilon \sigma_{a_t})$ is a diffeomorphism. Thus the symmetry map $\sigma_{a_t}^\epsilon$ i.e. Eq. (15) constitutes a local diffeomorphism. The above proof culminating in Eq. (24) implies that we have a local diffeomorphism mapping solutions of the ODEs to solutions, thus a symmetry of the system of ODEs.

**Limitations.**  However $D, E$ to be invertible is a strong assumption as it implies that the Koopman space approximation admit the same dimension as the state space. Note that however in practice as we only require $D \circ E \approx id$ one may choose other Koopman space dimensions.

**Local Lie group.**  What is left to show is that the symmetry of the system of ODEs locally is a Lie group. In definition 2 we need to show points (1)-(3).

1. For $\epsilon = 0$ one finds that $\sigma_{a_t}^0$ is the identity map i.e for such that

$$\tilde{s}_t(s_t, 0) = D\big((\mathbb{1} + 0 \cdot \sigma_{a_t})E(s_t)\big) = D(E(s_t)) = s_t \quad (28)$$

---

[6]For notational simplicity we study the general case $(\mathbb{1} + \epsilon \sigma_{a_t}) \to \sigma_{a_t}$.

2.

$$
\begin{aligned}
\sigma^{\epsilon_1}\sigma^{\epsilon_2} &= D\big((\mathbb{1}+\epsilon_1\sigma_{a_t})(\mathbb{1}+\epsilon_2\sigma_{a_t})E(s_t)\big) & (29)\\
&= D\big((\mathbb{1}+\epsilon_1\sigma_{a_t}+\epsilon_2\sigma_{a_t}+\mathcal{O}(\epsilon_1\cdot\epsilon_2))E(s_t)\big)\\
&= D\big((\mathbb{1}+(\epsilon_1+\epsilon_2)\sigma_{a_t}+\mathcal{O}(\epsilon_1\cdot\epsilon_2))E(s_t)\big)\\
&= \sigma^{\epsilon_1+\epsilon_2}
\end{aligned}
$$

for every $|\epsilon_1|,|\epsilon_2|\ll 1$.

3. $\tilde{s}_t(s_t,\epsilon)$ admits a Taylor series expansion in $\epsilon$, i.e. in a neighbourhood of $s$ determined by $\epsilon=0$. We may Taylor expand D around the point $E(s_t)\equiv g$ as

$$
\tilde{s}_t(s_t,\epsilon) = D(E(s_t)) + \epsilon\sum_{I=1}^{N}(\sigma_{a_t}E(s_t))_I\frac{\partial D}{\partial g_I} + \mathcal{O}(\epsilon^2) \tag{30}
$$

thus

$$
\tilde{s}_t(s_t,\epsilon) = s_t + \epsilon\zeta(s_t) + \mathcal{O}(\epsilon^2) \tag{31}
$$

for $\zeta(s_t) = \sum_{I=1}^{N}(\sigma_{a_t}E(s_t))_I\frac{\partial D}{\partial g_I}$.

4. The existence of an inverse element $(\sigma_{a_t}^{\epsilon})^{-1}=\sigma_{a_t}^{-\epsilon}$ follows from (1) and (2).

**Numerical errors.** Let us next turn to the second part of the theorem incorporating for practical numerical errors. We aim to show that under the assumption $[\sigma_{a_t}^{\epsilon},\mathcal{K}(a_t)]=\epsilon_a\mathbb{1}$ one finds $\tilde{s}_{t+1}=\tilde{F}(\tilde{s}_t,a_t)+\mathcal{O}(\epsilon_a)$. Note that $\sigma_{a_t}\to(\mathbb{1}+\epsilon\sigma_{a_t})$ admits an expansion in the parameter $\epsilon$. For $\epsilon\ll 1$ one can Taylor expand the differentiable functions $D,E$ to find that

$$
\tilde{s}_t = s_t + \delta(\epsilon)\,,\quad\text{with }\ \delta(\epsilon) = \epsilon\sum_I g_I^{\sigma}\frac{\partial D}{\partial g_I}\Big|_{E(s_t)} + \mathcal{O}(\epsilon^2)\,, \tag{32}
$$

where $g^{\sigma}=\sigma_{a_t}\hat{*}E(s_t)$ and $g=E(s_t)$, and with the index $I=1,\ldots,N$. The analog expression holds for $t+1$ i.e. $\tilde{s}_{t+1}=s_{t+1}+\mathcal{O}(\epsilon)$. If $a_t=a_t(s_t)$ is differentiable function of the states[7] one may Taylor expand the Koopman operator using equation 32 as[8]

$$
K(a_t(\tilde{s}_t)) = K(a_t(s_t)) + \sum_i\sum_n\delta_n(\epsilon)\,a'_{in}(s_t)\,\mathcal{K}_i = K(a_t(s_t)) + \epsilon\Delta + \mathcal{O}(\epsilon^2)\,. \tag{33}
$$

where and $a'_{in}(s_t)=\frac{\partial a_t}{\partial s_{t,n}}\big|_{s_t}$ and $n=1,\ldots,dim(s_t)$ and with

$$
\Delta = \sum_i\sum_n\sum_I g_I^{\sigma}\frac{\partial D_n}{\partial g_I}\Big|_{E(s_t)}a'_{in}(s_t)\,\mathcal{K}_i\,. \tag{34}
$$

However, by the definition of the symmetry map $a_t(\tilde{s}_t)=a_t(s_t)$, thus the Koopman operators in equation 33 match $\forall\epsilon$. By using the assumption on the error one infers from equation 26 that

$$
\begin{aligned}
\tilde{s}_{t+1} = D\Big((\mathbb{1}+\epsilon\sigma_{a_t})\hat{*}\mathcal{K}(a_t)E(s_t)\Big) = \quad & D\Big((\mathcal{K}(a_t)+\epsilon_a\mathbb{1})\,(\mathbb{1}+\epsilon\sigma_{a_t})\hat{*}E(s_t)\Big)\\
= \quad & D\Big(\mathcal{K}(a_t))\,(\mathbb{1}+\epsilon\sigma_{a_t})\hat{*}E(s_t)+\epsilon_a\,E(s_t)+\mathcal{O}(\epsilon_a\epsilon)\Big)\\
= \quad & D\Big(\mathcal{K}(a_t)E(\tilde{s}_t)+\epsilon_a\,E(s_t)+\mathcal{O}(\epsilon_a\epsilon)\Big)\,, \quad (35)
\end{aligned}
$$

One may Taylor expand $D$ by making use of $\epsilon_a\ll 1$ as

$$
\tilde{s}_{t+1} = D\big(\mathcal{K}(a_t)E(\tilde{s}_t)\big) + \epsilon_a\sum_I E_I(s_t)\frac{\partial D}{\partial g_I}\Big|_{\mathcal{K}(a_t)E(\tilde{s}_t)} + \mathcal{O}(\epsilon_a\epsilon) + \mathcal{O}(\epsilon_a^2)\,. \tag{36}
$$

---

[7]Although neural networks modeling the policies modeling the data-distribution contain non-differentiable components they are differentiable almost everywhere.

[8]The reader may wonder about the explicit use of the index $\delta_n(\epsilon)$ expressing the dimension of the state. it is necessary here although we have simply used $s_t$ as an implicit vector without explicitly specifying any components throughout the work.

Thus under the assumption that the numerical error is given by a violation of the commutation relation one finds $\tilde{s}_{t+1} = \tilde{F}(\tilde{s}_t, a_t) + \mathcal{O}(\epsilon_a)$ where

$$\epsilon_a \sum_I E_I(s_t) \frac{\partial D}{\partial g_I}\Big|_{\mathcal{K}(a_t)E(\tilde{s}_t)} \sim \mathcal{O}(\epsilon_a) \quad . \tag{37}$$

By comparison with equation 32 one infers that the $\epsilon$-expansion is a good approximation to the real dynamic if $\epsilon_a \ll \epsilon$. This concludes the proof of theorem 3.5. Let us note that the proof for state shifts arising from a sum of symmetry transformations as

$$\tilde{s}_t = D\Big( \big(\mathbb{1} + \sum_{I=1}^N \epsilon_I \, \sigma_{a_t}^I\big) \hat{\ast} E(s_t)\Big) , \quad \tilde{s}_{t+1} = D\Big( \big(\mathbb{1} + \sum_{I=1}^N \epsilon_I \, \sigma_{a_t}^I\big) \hat{\ast} E(s_{t+1})\Big) \quad . \tag{38}$$

is completely analogous, and an analogous theorem holds. We choose the number of symmetries to run over the dimension of the Koopman latent space N as it is relevant for the **KFC++** setup.

## C    IMPLEMENTATION DETAILS

**The Koopman forward model** The KFC algorithm requires pre-training of a Koopman forward model $\mathcal{F} : \mathcal{S} \times \mathcal{A} \to \mathcal{S}$ as

$$\mathcal{F}^c(s_t, a_t) = \begin{cases} D\big(E(s_t)\big) = s_t & \text{if } c = 0 \text{ corresponds to a VAE.} \\ D\Big( \big(\mathcal{K}_0 + \sum_{i=1}^m \mathcal{K}_i \, a_{t,i}\big) E(s_t) \Big) = s_{t+1} & \text{if } c = 1\text{: forward prediction model.} \end{cases} \tag{39}$$

where both of $E$ and $D$ are approximated by Multi-Layer-Perceptrons (MLP's) and the bilinear Koopman-space operator approximation are implemented by a single fully connected layer for $\mathcal{K}_{i=0,\dots,m}$, respectively. The model is trained on batches of the offline data-set tuples $(s_{t+1}, s_t, a_t)$ and optimized via the loss-function

$$\bar{l}_2(\mathcal{F}^1(s_t, a_t), \, s_{t+1}) \; + \; \gamma_2 \, \bar{l}_2(\mathcal{F}^0(s_t + s_\epsilon, a_t), \, s_t + s_\epsilon) \quad , \tag{40}$$

where $s_\epsilon$ is a normally distributed random sate vector shift with zero mean and a standard deviation of $6 \cdot 10^{-2}$ where the $\bar{l}_2$-loss is the Huberloss. For training the Koopman forward model in Eq. (16) we split the dataset in a randomly selected training and validation set with ratios $70\%/30\%$. We then loop over the dataset to generate the required symmetry transformations and extend the replay buffer dataset with the latter according to Eq. (18) as

$$\textbf{KFC} : \quad \big(s_t, s_{t+1}, a_t, r_t\big) \stackrel{symmetries}{\longmapsto} \big(s_t, s_{t+1}, a_t, r_t, \sigma_{a_t}\big) \quad , \tag{41}$$

$$\textbf{KFC++} : \quad \big(s_t, s_{t+1}, a_t, r_t\big) \stackrel{symmetries}{\longmapsto} \big(s_t, s_{t+1}, a_t, r_t, U(a_t), U(a_t)^{-1}\big) \quad .$$

It is more economic to compute the symmetry transformation as a pre-processing step rather than at runtime of the RL algorithm.

Let us reemphasize a couple of points addressed already in the main text to make this section self-contained. Note that we have built upon the implementation of CQL (Kumar et al., 2020), which is based on SAC (Haarnoja et al., 2018). **We also use the same hyperparameters for all the experiments as presented in the CQL and S4RL papers for baselines.**

**Koopman forward VAE-model:** For the encoder as well as the decoder we choose a three layer MLP architecture, with hidden-dimensions $[512, 512, N]$ and $[512, 512, m]$, respectively. Where $N = 32$ is the Koopman latent space dimension and $m$ is the state space dimensions following our notation from section 2. The activation functions are ReLU (Nair & Hinton, 2010). We use an ADAM optimizer with learning rate $3 \cdot 10^{-4}$. We train the model for 75 epochs. The batch size is chosen to be 256. Lastly, we employ a random normally distributed state-augmentation for the VAE training i.e. $s \to s + s_\epsilon$ with $s_\epsilon \in \mathcal{N}(0, 1 \cdot 10^{-2})$.

**Hyper-parameters KFC:** We take over the hyper-parameter settings from the CQL paper (Kumar et al., 2020) except for the fact that we do not use automatic entropy tuning of the policy optimisation step Eq. (3) but instead a fixed value of $\alpha = 0.2$. The remaining hyper-parameters of the conservative

Q-learning algorithm are as follows: $\gamma = 0.99$ and $\tau = 5 \cdot 10^{-3}$ for the discount factor and target smoothing coefficient, respectively. Moreover, the policy learning rate is $1 \cdot 10^{-4}$ and the value function learning rate is $3 \cdot 10^{-4}$ for the ADAM optimizers. We enable Lagrange training for $\tilde{\alpha}$ with threshold of 10.0 and learning rate of $3 \cdot 10^{-4}$ where the optimisation is done by an ADAM optimizer. The minimum Q-weight is set to 10.0 and the number of random samples of the Q-minimizer is 10. For more specifics on these quantities see (Kumar et al., 2020). The model architectures of the policy as well as the value function are three layer MLP's with hidden dimension 256 and $ReLU$-activation. The batch size is chosen to be 256. Moreover, the algorithm performs behavioral cloning for the first $40k$ training-steps, i.e. time-steps in an online RL notation.

The KFC specific choices are as follows: the split of random to Koopman-symmetry state shifts is 20/80 i.e. $p_K = 80\%$, whereas the option for the symmetry-type is either "Eigenspace" or "Sylvester" referring to case (I) and (II) of Eq. (18), respectively. Regarding the random variables in the symmetry transformation generation process, we choose $\epsilon_i \in \mathcal{N}(0, 1 \cdot 10^{-4})$ for case (II) and $\epsilon \in \mathcal{N}(0, 5 \cdot 10^{-5})$ for case (I) in Eq. (18) after normalizing $\sigma_a$ by its mean. While for the random shift we use $\tilde{\epsilon} \in \mathcal{N}(0, 3 \cdot 10^{-3})$ which is the hyper-parameter used in (Sinha et al., 2021).

**Computational setup:** We performed the empirical experiments on a system with PyTorch 1.9.0a (Paszke et al., 2019) The hardware was as follows: NVIDIA DGX-2 with 16 V100 GPUs and 96 cores of Intel(R) Xeon(R) Platinum 8168 CPUs and NVIDIA DGX-1 with 8 A100 GPUs with 80 cores of Intel(R) Xeon(R) E5-2698 v4 CPUs. The models are trained on a single V100 or A100 GPU.

# D   ABLATION STUDY

| Task Name | S4RL-($\mathcal{N}$) | KFC | KFC++ | KFC++-contact |
|---|---|---|---|---|
| cheetah-random | **52.3** | 48.6 | 49.2 | 49.4 |
| cheetah-medium | 48.8 | 55.9 | 59.1 | **61.4** |
| cheetah-medium-replay | 51.4 | **58.1** | **58.9** | 59.3 |
| cheetah-medium-expert | **79.0** | **79.9** | **79.8** | **79.8** |
| hopper-random | **10.8** | 10.4 | 10.7 | 10.7 |
| hopper-medium | 78.9 | 90.6 | **94.2** | **95.0** |
| hopper-medium-replay | 35.4 | **48.6** | **49.0** | 49.1 |
| hopper-medium-expert | 113.5 | 121.0 | 125.5 | **129** |
| walker-random | **24.9** | 19.1 | 17.6 | 18.3 |
| walker-medium | 93.6 | 102.1 | **108.0** | 108.3 |
| walker-medium-replay | 30.3 | **48.0** | 46.1 | 48.5 |
| walker-medium-expert | 112.2 | 114.0 | **115** | 118.1 |

**Table 2:** We study the effect of combining S4RL-$\mathcal{N}$ based augmentation training on "contact" event transitions (state transitions where the agent makes contact with a surface, read D.1 for more details), and KFC++ augmentation training on non-"contact" events. We see that KFC++-contact performs similarly and in most cases, slightly better than the KFC++ baseline, which is expected. We experiment with the Open AI gym subset of the D4RL tasks and report the mean normalized returns over 5 random seeds.

## D.1   TRAINING ON NON CONTACT EVENTS

To test the theoretical limitations in a practical setting one interesting experiment is to use the S4RL-$\mathcal{N}$ augmentation scheme when the current state is a "contact" state and to choose KFC++ otherwise. We define a "contact" state as one where the agent makes contact with another surface; in practice we looked at the state dimensions which characterize the velocity of the agent parts, and if there is a sign difference between $s_t$ and $s_{t+1}$, then contact with a surface must have been made. The details regarding which dimension maps to agent limb velocities is available in the official implementation of the Open AI Gym suite (Brockman et al., 2016).

Note that the theoretical limitations are twofold, firstly the theorems 3.4 and 3.5 only hold for dynamical systems with differentiable state-transitions; secondly, we employ a Bilinearisation Ansatz for the Koopman operator. Contact events lead to non-continuous state transitions and moreover

would require incorporating for external forces in our Koopman description, thus both theoretical assumptions are practically violated in our main empirical evaluations of KFC and KFC++. By performing this ablation study which is more suitable for our Ansatz, i.e. the KFC++ part is only trained on "differentiable" trajectories without contact events. In other words, the Koopman description is unlikely to give good information regarding $s_{t+1}$ if there is a "contact event", we simply use S4RL-$\mathcal{N}$ when training the policies, which we expect to give slightly better performance than the KFC++ baseline.

We report the results in Table 2, where we perform results on the OpenAI Gym subset of the D4RL benchmark test suite. We denote the proposed scheme as **KFC++-contact** and compare to both S4RL-$\mathcal{N}$ and KFC++. We see that KFC++-contact performs comparably and slightly better in some instances, compared to KFC++. This is expected as contact events may be rare, or the data augmentation given by the KFC++ model during contact events is pseudo-random and itself is normally distributed, which may suggest that KFC++ behaves similarly to KFC++-contact. However, we do not say this conclusively.

## D.2 MAGNITUDE OF KFC AUGMENTATION

To further try to understand the effect of using KFC for guiding data augmentation, we measure the mean L2-norm of the strength of the augmentation during training. We measured the mean L2-norm between the original state and the augmented state, and found the distance to be $2.6 \times 10^{-4}$. For comparison, S4RL uses a fixed magnitude of $1 \times 10^{-4}$ (Sinha et al., 2021) for all experiments with S4RL-Adv. Since S4RL-Adv does a gradient ascent step towards the direction with the highest change, its reasonable for the magnitude of the step to be smaller in comparison. An adversarial step too large may be detrimental to training as the step can be too strong. KFC minimizes the need for making such a hyperparameter search decision.

## D.3 SYMMETRIES

In this section we provide a quantitative analysis of the symmetry shifts generated by KFC++ in comparison to random shifts of the latent states. For the latter we choose the variance such that absolute state shift

$$\Delta S := |\tilde{s}_t - s_t| + |\tilde{s}_{t+1} - s_{t+1}| \tag{42}$$

averaged over all sampled data points (D4RL) is comparable between the two setups. Moreover we simply use the encoder We then compare how well the shifted states are in alignment with the actual dynamic using the online Mujoco environment .[9] To do so we set the environment state to the value $\tilde{s}_t$ and use the action $a_t$ to perform the environment step which we denote by $s_{t+1} = M(s_t, a_t)$[10] The performance metric is then given by

$$\Delta E := |s_{t+1} - M(\tilde{s}_t, a_t)| \ . \tag{43}$$

Note that $\Delta E$ is simply the error to the actual dynamic of the environment. We take the model trained on the hopper-medium-expert dataset from section appendix D.1 to compute the symmetry shift. Moreover, for the comparison in equation 43 we split the state space into positions and velocities of the state vector, i.e. the first five elements of $s_{t+1} - M(\tilde{s}_t, a_t)$ give position while the remaining ones given velocities. The norm is taken subsequently. For the results see figure D.3.

We conclude that there is a qualitative difference between the distributions obtained. Most notable the symmetry shift allows for a controlled shift by a larger $\Delta S$ while still maintaining a small error $\Delta E$. This amounts to a wide but accurate exploration of the environments state-space. Although we cannot say this conclusively we expect this to be the reason for the performance gains of KFC++ over random shifts.

## D.4 PREDICTION MODEL

As a further ablation study it is of interest to compare the symmetry induced state shifts tuple $(\tilde{s}_t, \tilde{s}_{t+1})$ to one obtained by the forward prediction of our VAE-forward model equation 16.

---

[9]Note that this requires a minor modification of the environment code by a function which allows to set states to specific values.

[10]$M(s_t, a_t)$ denotes the step done by the active gym environment.

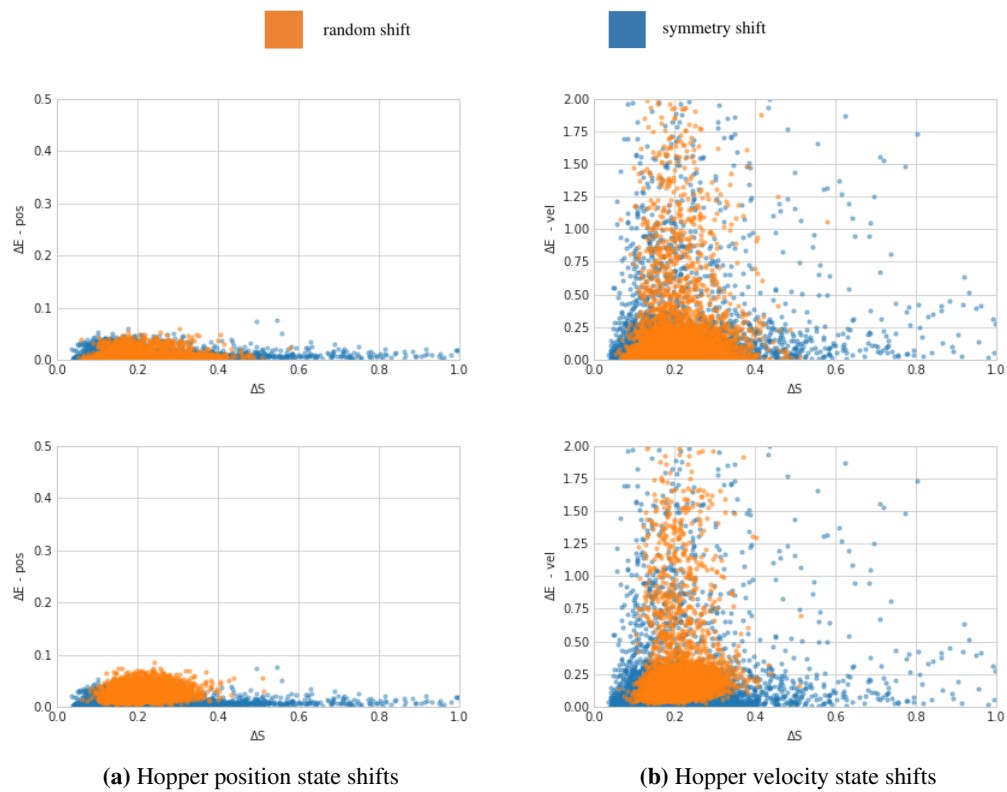

**(a)** Hopper position state shifts   **(b)** Hopper velocity state shifts

**Figure 3:** Symmetry shifts vs. random latent space shifts of same magnitude in the mean of $\Delta S$ compared by its accuracy in the online evaluation in the gym Hopper-v3 environment. The first line shows the result where $s_t$ and $s_{t+1}$ are shifted by the same N-dimensional random variable. For the ablation study in line two we sample twice from the random variable for the shifts $s_t$ of $s_{t+1}$, respectively.

**KFC++prediction:** In particular we use the forward prediction on $\tilde{s}_t$ obtained by the KFC++ shift, i.e. $\tilde{s}_{t+1} = \mathcal{F}^{c=1}(\tilde{s}_t, a_t)$. We refer to this setup in the following as KFC++prediction. See table 3 for the results.

**Fwd-prediction:** Moreover, we study the case where we $\tilde{s}_t$ is obtained by a shift with a normally distributed random variable $\mathcal{N}(0, 6 \cdot 10^{-3})$ and then simply forward predict as $\tilde{s}_{t+1} = \mathcal{F}^{c=1}(\tilde{s}_t, a_t)$. This is comparable to conventional model based approaches employing our simple VAE forward model. This constitutes an ideal systematic comparison as the model, hyper-parameters and training procedure are identical to the one used in the KFC variants. Moreover, the variance of the random variable is such that the distance to the augmented states is comparable to the ones in KFC, see appendix D.2. See table 4 for the results.

We conclude that **KFC++prediction** falls behind both **KFC** as well as **KFC++**. This was expected as the symmetry shift in the latter alters the original data points by means of the VAE which admits not only a much higher accuracy but also advanced generalisation capabilities to out of distribution values.

More interesting however is that **Fwd-prediction** falls behind both **KFC** as well as **KFC++** as well as **KFC++prediction**. This is a strong indicator that the symmetry shifts provide superior out-of-distribution data for training a Q-learning algorithm.

## D.5 DISCUSSION

Although our KFC framework is suited best for the description of environments described by control dynamical systems the learned (in self-supervised manner) linear Koopman latent space representation may be applicable to a much wider set of tasks. However, there are notable shortcomings to the current implementation both conceptually as well as practically. The bilinearisation in Eq. (16) of the latent space theoretically assumes the dynamical system to be governed by Eq. (8), which is rather restrictive. Although a general non-affine control system admits a bilinearisation (Brunton et al., 2021)

| Task Name | KFC | KFC++ | KFC++-prediction |
|---|---|---|---|
| cheetah-random | 48.6 | **49.2** | 46.5 |
| cheetah-medium | 55.9 | **59.1** | 53.7 |
| cheetah-medium-replay | **58.1** | **58.9** | 55.3 |
| cheetah-medium-expert | **79.9** | **79.8** | 76.3 |
| hopper-random | **10.4** | **10.7** | **10.8** |
| hopper-medium | 90.6 | **94.2** | 90.5 |
| hopper-medium-replay | **48.6** | **49.0** | 44.2 |
| hopper-medium-expert | 121.0 | **125.5** | 121.2 |
| walker-random | **19.1** | 17.6 | 15.6 |
| walker-medium | 102.1 | **108.0** | 105.3 |
| walker-medium-replay | **48.0** | 46.1 | 45.2 |
| walker-medium-expert | 114.0 | **115.6** | 114.5 |

**Table 3:** Results with prediction model KFC++-prediction on the Open AI Gym subset of the D4RL tasks. We report the mean normalized episodic rewards over 5 random seeds similar to the original D4RL paper Fu et al. (2021).

| Domain | Task Name | KFC | KFC++ | Fwd-prediction |
|---|---|---|---|---|
| AntMaze | antmaze-umaze | **96.9** | **99.8** | 92.7 |
| | antmaze-umaze-diverse | **91.2** | **91.1** | 90.1 |
| | antmaze-medium-play | 60.0 | **63.1** | 60.8 |
| | antmaze-medium-diverse | 87.1 | **90.5** | 88.0 |
| | antmaze-large-play | 24.8 | **25.6** | 23.1 |
| | antmaze-large-diverse | **33.1** | **34.0** | 29.3 |
| Gym | cheetah-random | 48.6 | 49.2 | **50** |
| | cheetah-medium | 55.9 | **59.1** | 50.1 |
| | cheetah-medium-replay | **58.1** | **58.9** | 56.4 |
| | cheetah-medium-expert | **79.9** | **79.8** | 71.3 |
| | hopper-random | **10.4** | **10.7** | **10.4** |
| | hopper-medium | 90.6 | **94.2** | 82.3 |
| | hopper-medium-replay | **48.6** | **49.0** | 40.8 |
| | hopper-medium-expert | 121.0 | **125.5** | 120.3 |
| | walker-random | **19.1** | 17.6 | 18.4 |
| | walker-medium | 102.1 | **108.0** | 103.2 |
| | walker-medium-replay | **48.0** | 46.1 | 41.7 |
| | walker-medium-expert | 114.0 | **115.3** | 111.8 |
| Adroit | pen-human | **61.3** | 60.0 | 49.4 |
| | pen-cloned | **71.3** | 68.4 | 50.2 |
| | hammer-human | 7.0 | **9.4** | 6.1 |
| | hammer-cloned | 3.0 | **4.2** | 4.2 |
| | door-human | 44.1 | **46.1** | 41.8 |
| | door-cloned | 3.6 | **5.6** | 1.2 |
| | relocate-human | **0.2** | **0.2** | 0.2 |
| | relocate-cloned | **-0.1** | **-0.1** | **-0.1** |
| Franka | kitchen-complete | **94.1** | **94.9** | 90.0 |
| | kitchen-partial | 92.3 | **95.9** | 84.6 |

**Table 4:** Results with prediction model Fwd-prediction on the D4RL tasks. We report the mean normalized episodic rewards over 5 random seeds similar to the original D4RL paper Fu et al. (2021).

it generically requires the observables to depend on the action-space variables implicitly. Secondly, the Koopman operator formalism is theoretically defined by its action on an infinite dimensional

observable space. The finite-dimensional approximation i.e. the latent space representation of the Koopman forward model in Eq. (16) lacks accuracy due to that. On the practical side our formalism requires data pre-processing which is computationally expensive, i.e. solving the Sylvester or eigenvalue problem for every data-point. Moreover, the Koopman forward model in Eq. (16) serves as a function approximation to two distinct tasks. Thus one faces a twofold accuracy vs. over-estimation problem which needs to be balanced. The systematic error in the VAE directly imprints itself on the state-data shift in Eqs. (15) and (18) and may thus conceal any potential benefit of the symmetry considerations. Lastly, the dynamical symmetries do not infer the reward. Thus the underlying working assumption is that the reward should not vary much in Eq. (15).

**A note on the simplicity of the current algorithm:** Let us stress a crucial point. Algorithmically the symmetry maps are derived in two distinct ways **KFC** and **KFC++**. The latter, constitute a simple starting point to extract symmetries from our setup. More elaborate studies employing the extended literate on Koopman spectral analysis are desirable. Moreover, it is desirable to extend our framework to more complex latent space descriptions such as e.g. world models. Both on a theoretical as well as a practical level. It is our opinion that by doing so there is significant room for improvement both in the accuracy of the derived symmetry transformations as well their induced performance gains of Q-learning algorithms. Note that currently our VAE model is of very simple nature and the symmetries are extracted in a rather uneducated way. While the Sylvester algorithm simply converges to one out of many symmetry transformations for the KFC++ algorithm we omit all the information of the imaginary part, let alone utilize concrete spectral information.

