# OpenReview forum: "Koopman Q-learning: Offline Reinforcement Learning via Symmetries of Dynamics"
_ICLR.cc/2022/Conference — ICLR 2022 Submitted_

### Official Review · Reviewer_ohJ3 · 2021-10-31

**Correctness:** 3
**Technical Novelty And Significance:** 2
**Empirical Novelty And Significance:** 2
**Recommendation:** 5
**Confidence:** 3

**Main Review:**


### Strengths
- The idea of using learned symmetries of the environment transition operator to do data augmentation in latent space is an intriguing one and to my knowledge is novel.

- The work provides sufficient background that a reader with a background in RL and with a passing familiarity with linear dynamical systems could understand the methodology and goal of the paper.

- I didn’t notice any obvious mistakes in the proofs of the main theorems, though there were several typos (e.g. $ \bar{\sigma} *$ in place of $\sigma \bar{*}$).

- Empirically, the method does seem to outperform the baselines the authors chose.

Weaknesses:
- The paper suffers from **lack of clarity**: some symbols are not defined and the types of certain functions and operators must be inferred by referring back to their initial definition. For example, the notation U(a)g used in equation 13 is ambiguous at first as to whether it is meant to refer to a composition of operators or to multiplication of the outputs.
    - In lemma 3.3, $\bar{\Sigma}$ is not defined — further, using the most recent definition of the group would lead me to believe it acts on state-action pairs, however in Lemma 3.3 it is applied to individual states.
    - The paper states that epsilon is sampled from normal distribution, but doesn’t state the precise variance, only claiming epsilon <<1.
    - The operator $\sigma^\epsilon_{a_t}$ is not defined in Theorem 3.7, nor is it clear how it relates to the (presumed) group element $\sigma$.
    - The notation $\mathcal{F}(S \times A)$ is not defined. Further, the type/definition of U(a) doesn’t make sense to me: what is the value of (U(a)(g))(s’, a’) ?
    - the notation $\bar{\Sigma}$, $\hat{\Sigma}$, etc. should be formally defined. Further, in many definitions the notations $\phi$ and $*$ are used interchangeably.


- There is a **large gap between the assumptions used in the theoretical results and the empirical setting**, both in terms of the assumptions required and in the outcome being measured. The theoretical results concern the equivariance properties of the Koopman operator, while the empirical results concern the performance of a learned model.
    - The proposed method looks suspiciously similar to a VAE objective where the perturbation is defined by sigma. Thus it is not clear to me whether the properties of the learned transition model + invariances are actually important to the proposed success of the method, or indeed whether the learned symmetries can really be said to be true symmetries of the Koopman operator as opposed to simply latent-space perturbations
    - The existence of the family of operators U(a) used in Lemma 3.5 is not shown in main paper, and indeed additional conditions are discussed in the appendix in order for these operators to exist. It is not shown that these conditions are satisfied in the settings the paper is interested in.
    - It is not clear what types of symmetries exist in the evaluation environments, or whether KFC is able to learn a model that is equivariant to them.
    - The jump from the theoretical results on equivariance to the empirical evaluation on performance is quite abrupt and it is not clear how the theoretical results are meant to indicate that the method will indeed improve performance.
    - It’s not clear how the learned encoder/decoder/observables used in practice satisfy the assumptions laid out in the theoretical section.


- **The empirical results are not convincing.** The observation that S4RL outperforms KFC on the -random split where model-learning is less useful suggests that most of the benefit of KFC is coming from the model learning component of the algorithm. S4RL and CQL are both model-free methods, so this seems to suggest to me that it might not be the learned latent-space symmetries but rather the model-learning auxiliary task that is making KFC outperform the selected baselines. Additionally,  a number of algorithmic details are missing from the description of the method, making it difficult to interpret the experimental results.
    - I couldn’t find a description of KFC++ anywhere in the paper.
    - How is the matrix representation of the koopman operator obtained?
    - How many training steps do the methods get? Does the model-training phase count towards the  KFC method’s “optimization budget”?

### Potential Improvements


- Evaluation against stronger baselines that incorporate model learning and equivariance in the latent space (e.g. DeepMDP, DBC).

- Rewriting the paper to improve the clarity: making sure the types of functions and operators are always clear, using consistent notation, explicitly stating the necessary assumptions for the theoretical results to hold, etc.

- Using more targeted experiments to study the learned symmetries of the model will provide greater insight into whether the theoretical results hold in the latent space of KFC agents.


### Notes
-  I didn’t understand the claim “an exploration of the environment’s phase space” (section 1, page 2)
- In environments that are not equivariant to any symmetry group, how would you expect the method to behave?
- There are a number of typos in the manuscript: “the existence on nature” on p2, ’raod-map’ on p6, “regulizer” on pp 3 & 6, “in a the free” just after eq 8, etc. These were separate from my concerns about clarity.

**Summary Of The Paper:**

This paper aims to improve generalization and data efficiency in offline RL using a latent-space data augmentation approach inspired by results from Koopman operator theory. The resulting method is evaluated on several environments against a similar method which uses a form of self supervision via data augmentation.

**Summary Of The Review:**

Overall, my concerns over the clarity of the paper, the gap between theory and experimental results, and the limited comparison to existing methods lead me to recommend that the paper be rejected.

---

> ### Author Response · Authors · 2021-11-15
> **Response to Reviewer ohJ3 - Part 1**
>
> We would like express our gratitude for your detailed, critical, and constructive feedback.
> Let us address your concerns in the following.
>
>
> > **I:** Regarding your point *"The paper suffers from lack of clarity..."*. We are working on in improvement of the clarity of the theory section for the revised version currently.
>
> * **1:** It does act on state-action pairs, where actions are subject to the trivial group, i.e. a group with only the identity element as an element . We do not believe that there is any contraction or ambiguity in between the definition and Lemma (3.3).
> * **2:** We do discuss this point in the hyper-parameter section in the appendix, please see Appendix B, hyper-parameters KFC, last paragraph.
> * **3:** It states in Theorem 3.7 that that
>     the system is of the form as in Theorem 3.6 where it is given that $\sigma_{a_t}$ commutes with the Koopman operator, the map $\sigma_{a_t}^\epsilon$ is defined in eq.(17). We agree that the connection to the symmetry group needs to expressed more clearly in a revised version.
> * **4:**
>     $\mathscr{F}(\mathcal{S} \times \mathcal{A})$ is defined as the (Banach) space of eigenfunctions in the paragraph after eq.(12).
> * **5:**
>     Thanks for that note, we will get rid of $\phi, \bar\phi$.
>      Also we will include more formal definitions in the revised version. Some of which however will need to be in the appendix.
>
> Concludingly, a majority of your points is indeed addressed in our initial submission.
>
> > **II** Regarding your second major criticism *"There is a large gap between the assumptions used in the theoretical results and the empirical setting, both in terms of the assumptions required and in the outcome being measured..."*
>
>
> * **1:** We believe we address this point in our existing ablation study appendix C.3. An extension will be in the revised version. The fact that an operator acting on the Koopman space commutes with the Koopman operator guarantees that it maps solution  of the underlying system of differential equations to each other. In the non-control setting this simple translates to a mapping of solutions with different initial conditions. It is a symmetry in that regard.
> \
> In general one does not know if the Koopman operator in a simple  VAE model is a good approximation to the real Koopman operator of the system. However, one aims to find a Koopman operator which models the data distribution i.e. a system of coupled linear ODEs which describe the data. The computed symmetries are of the latter approximate system.
> * **2,3, and 5:** We do discuss related points e.g. we explicitly stress that some of our assumption are not satisfied in the settings we are interested in empirically and even perform an ablations study with appendix C.1 with that sole focus. The particular points you repeatedly raise regarding if our symmetries connect true solutions of the system we have addressed this in point (1). Moreover, we conclude from ablation study C.1 that even in the cases where our assumptions are explicitly violated our approach works well i.e. in all environments with contact events.
> * **4:** Could you please provide specific examples of improvements you have in mind?
>    We provide a data augmentation framework. To our knowledge  theoretical results which guarantee that all sorts of data-augmentation must lead to a better performance are very rare. Thus the jump you are referring to seems to be intrinsic in the nature of our work.
>    However, we agree that there are some improvements to be made on the level of presentation rather then the main content of our theoretical contribution e.g. to empirically analyse a simpler RL environment as a toy example to ease the transition ( probably added to the appendix ).

---

> > ### Author Response · Authors · 2021-11-15
> > **Response to Reviewer ohJ3 - Part 2**
> >
> > > **III:**  *"The empirical results are not convincing."*
> >
> > *  *"The observation that S4RL outperforms KFC on the -random split where model-learning is less useful suggests that most of the benefit of KFC is coming from the model learning component of the algorithm."*
> > \
> > Our ablation study C.3 would suggest that your conclusion is incorrect. In fact KFC++ outperforms "KFC++ prediction" also on the random sets. Thus a more likely explanation is that our simple VAE model cannot learn the dynamics reasonably well on the random sets. The latter is also supported by the fact that for the simplest environment dynamics i.e. the Hooper environment there are no differences in performance, neither for the ablation study nor for our main results. Which makes us confident that our conclusion here is accurate.
> > However, we will nevertheless study this point in more detail in the revised version.
> >
> > *  *"S4RL and CQL are both model-free methods, so this seems to suggest to me that it might not be the learned latent-space symmetries but rather the model-learning auxiliary task that is making KFC outperform the selected baselines."*
> > \
> > We would refer again to ablation study C.3.  KFC++ outperforms "KFC++ prediction".
> > * **1:** We apologize for that. Algorithm (I) in eq(19) is KFC and (II) is KFC++. We have changed equation (19) in the revision to reflect this directly. Fixed in revised version.
> > * **2:**
> >     We do provide this after eq.(21): "the Koopman-space operator approximation are implemented by a single fully connected layer...".
> > * **3:**
> >     The only difference between KFC and CQL or S4RL is that it requires training a model before we start learning the value functions.
> >      Note that the same trained model then can be used for Hyper-parameter search for the Q-learning algorithm or for employed on different Q-learning algorithms.  In that light the computational resources spent on model training compared to Q-learning is negligible.  Moreover, our framework is to be seen as a data augmentation scheme. One may provide new datasets including its symmetry transformations  i.e. (dataset, symmetries transformations) available for download.
> >      However, we are not sure what the reviewers means with "optimization budget".
> >
> > > **IV:** *"Potential Improvements"*
> >
> > * **1:** Thank you for this suggestion. As mentioned above, we do provide an ablation study against a strong  model based baseline "KFC++ prediction" in appendix C.3. We are working on extending this study. We are considering replacing one of the S4RL baselines in the main text.
> > \
> > However, we do not agree with your suggestion that the above references would constitute appropriate baselines. Firstly, our model for  future sate prediction is a simple MLP processing only one time step. Secondly, we do not model the reward. Both of this points are certainly interesting directions to extend our framework. E.g. when applied to "world models"  with complex latent space dynamic our entire approach would need to be adjusted to be on the same footing of complexity as such a model as baseline. Thus, this is an entirely new  work of research, and can lead to interesting avenues of research in the future.
> > * **2:** Work on improvement is in progress. However, a non-negligible fraction of your comments is already addressed in our initial submission, as pointed out above.
> > * **3:** Thank you for this suggestion. We agree that such a study will improve the paper. In progress.

---

> ### Author Response · Authors · 2021-11-23
> **Manuscript Update and request for discussion.**
>
> We hope to have answered your concerns in our rebuttal.
> Additionally, we have also updated the manuscript to improve comprehension as you had suggested.
>
> While further manuscript changes are not allowed at this point, but we would appreciate discussion about any remaining concerns.
> We would be happy to incorporate suggestions and clarifications in the next version of manuscript.

---

> > ### Comment · Reviewer_ohJ3 · 2021-11-30
> > **Most concerns addressed but disagree about baselines**
> >
> > Thanks to the authors for their detailed response. I appreciate the pointer to the ablations in the appendix, and I think the proposed revisions to the text will improve the paper. I disagree with the claim that DBC and DeepMDP would not be relevant baselines, as neither learns a model of the environment and, while reward is included as part of the representation-learning objective both methods otherwise to exhibit a similar spirit to KFC in their effort to encourage invariance to transformations that preserve the structure of the MDP.
> >
> > Overall, I do like the idea of this paper. I think studying symmetries in the latent space is a potentially powerful tool that could complement existing work in the literature which has shown that latent space models can be used for planning and improving representation learning. I think the baselines the work compares against are not ideal for deriving insight into the proposed method, and would like to see some justification for whether the learned approximate koopman operator is accurate enough for the learned symmetries to be meaningful. At the moment I am leaving my score, but will review the appendix in greater detail later and may update if I am convinced that method does indeed learn meaningful symmetries in the environment.

---

### Official Review · Reviewer_SsNL · 2021-11-02

**Correctness:** 3
**Technical Novelty And Significance:** 3
**Empirical Novelty And Significance:** 2
**Recommendation:** 6
**Confidence:** 4

**Main Review:**

-The general idea and the results of this work are interesting and seem to be novel.

-However, I found the paper to be rather hard to understand. Some important assumptions are not so clear and require more clarification. Also, some points and notations should be better explained.


**The main issues that, in my view, could improve the paper are:**

First, there are some grammatical mistakes as well as missing words and typos, for instance
- Page 5, line 23 and page 13, line 29:    .Which ---> which
- On page 4, line 9 in the proof of Theorem 3.6, the sentence "First of all note that by the definition the Koopman operator it obeys" is grammatically incorrect
- Page 16, line 1:  us ---> use
- Page 16, the second line below eq. (27): an controllable ---> a controllable

Second, some important assumptions are not so clear and require more clarification:
- On page 4, line 6, it is not clear what are f_i functions. Also, considering  "f(s,a)=f_0(s) + \Sum f_i (s) a_i ", it is not clear how eq. (7) can be written as eq. (8). Moreover, writing "f_i=0, ..., m" is a bit confusing.

There are also some comments on Theorems and their proofs:
- In Theorem 3.4, eq. (12), the first term is "g(s_{t+1})" or "g(s_{t+1}, a_{t+1})"? Also, in this theorem it is not clear under which conditions we can obtain eq. (12). So, more clarification in this regard is needed (this is also needed for eq. (14) in Theorem 3.6).
- In the proof of Lemma 3.3, only  "σ_0∈ ∑" is the identity element but not "g", although it is written "Let σ_0 and g be the identity element". This should also be considered in the proof of Lemma 3.5, i.e. only  "σ_0∈ \bar{∑}" is the identity element.
- In the proof of Theorem 3.4, line 7, "σ^" should be changed into "σ".
- In the proof of Theorem 3.6, line 4, what is "a"? Is it "a_t"? Also the first term is "g(s_{t+1})" or "g(s_{t+1}, a_{t+1})"?
- In the proof of Theorem 3.7, line 5, "σ_{ε}^{a_t}" should be better explained and needs to be clarified.


**Summary Of The Paper:**

The paper proposes a symmetry-based data augmentation technique derived from a Koopman latent space representation based on some theoretical results on symmetries of dynamical control systems and symmetry shifts of data. The authors empirically evaluated their method on several benchmark offline reinforcement learning tasks D4RL, Metaworld and Robosuite and showed that their framework consistently improves the state-of-the-art of Q-learning algorithms.

**Summary Of The Review:**

This paper needs to be written in a more precise way. Especially, some important assumptions are not so clear and require more clarification.

---

> ### Author Response · Authors · 2021-11-15
> **Response to Reviewer SsNL**
>
> We would like express our gratitude for your detailed and critical feedback.
> Regrading your main objection *"some important assumptions are not so clear and require more clarification"* let us address your two points mentioned
>
> * **5:**  *"it is not clear how eq.\,(7) can be written as eq. (8)".*  \
> The superficial answer to this is for dynamical system where the r.h.s of the defining differential equations eq.(5) is given by a Bilinearisation i.e. $
> f(s, a) = f_{0}(s) + \sum_{i}^{m}f_{i}(s) a_i
> $ for  $f_i,\, i=0,\dots,m$ i.e. $\mathcal{C}^1$-differentiable-vector valued functions. We agree that it is desirable to explain this results in Koopman theory  more extensively.
>     We have added a paragraph to the appendix explaining this point in some more detail. We apologize that our introduction to Koopman theory in the main text cannot be self-contained.  *"On page 4, line 6, it is not clear what are $f_i$ functions."*
> \
> We have explicitly added the information that $f_i,\, i=0,\dots,m$ are $\mathcal{C}^1$-differentiable-vector valued functions. Which however, was implicit in the original submission as it was specified a few lines before that $f(s,a)$ is a $\mathcal{C}^1$-differentiable-vector valued function.
>
> *  **6:**
> \
>  Eq.(12) does not belong to Theorem 3.4 but to the main text. We do not quite understand  your point here. However, our guess is that this connects to the point above which we have  already addressed.
> \
> The second point, when one may express a system as in Theorem 3.6. This is indeed a very good question, to which however there is not much more to say than that all Bi-linear systems eq.(8) are in this category. Otherwise one needs to check case by case if that is true, i.e. for systems where analytical solutions exist. Thus we believe that our discussion in the initial submission in that regard is comprehensive (although short).
> \
> On a related note. In practice one makes an Ansatz for the Koopman operator in  a model. It is hard to tell if it is a good approximation to the real Koopman operator of the system. However, the latter is partially irrelevant as one aims to find a Koopman operator which models the data distribution i.e. a system of coupled linear ODEs which describe the data.  This is supported by our ablation study in appendix C.1.
>
> **Minor:**
>
> *  **1-4,6,7,8:**
> \
>     Many thanks for pointing those out! We have fixed the typos and  wording issues in the proofs as well as main text you mentioned.
> * **9:** *"In the proof of Theorem 3.6, line 4, what is "a"? Is it $a_t$? Also the first term is $g(s_{t+1})$ or $g(s_{t+1}, a_{t+1})$?*
> \
> We have added a note in the proof that for notational simplicity we drop the subscripts referring to time in the remainder of this proof i.e.\, $s_t \to s$  and $a_t \to a$. The first term is $g(s_{t+1})$ as stated.
> *  **10:**
> \
>  In the initial submission the proof contained an intentional abuse of notation. We have specified more clearly in the revised version. Regarding  your specific point, we explicitly define the map $\sigma^{\epsilon}_{a_t}$ in eq.(16) and eq.(17).
> Note that all information was discussed in Theorem 3.6 and the definition of $\hat\Sigma$-action-equivariant control systems. We are revising this section currently, e.g.  we have improved this theorem to hold more generally.
>
>
> We believe this addresses many of your points directly.
> Other than those, we agree with you that more mathematical details should be added in a revised version to improve the clarity regarding  certain aspects.

---

> > ### Comment · Reviewer_SsNL · 2021-11-16
> > **My concerns are addressed well**
> >
> > I really appreciate the authors’ detailed reply! They address my concerns. I have updated my score to reflect this.

---

> > > ### Author Response · Authors · 2021-11-17
> > > **Thank you**
> > >
> > > We are glad to hear that our explanations were helpful and we really appreciate your reconsideration of the score. Thank you!

---

### Official Review · Reviewer_VgB7 · 2021-11-06

**Correctness:** 3
**Technical Novelty And Significance:** 3
**Empirical Novelty And Significance:** 2
**Recommendation:** 6
**Confidence:** 3

**Main Review:**

Strengths:
1. The paper provides a principled way to generate data augmentation for offline RL (and for dynamical systems in general)
2. The paper improves over past data augmentation methods (S4RL variants) and show improved performance on D4RL tasks, meta world and robosuite environments

Concerns:
1. The paper claims "Current algorithms over-fit to the training dataset and as a consequence perform poorly when deployed
to out-of-distribution generalizations of the environment. We aim to address these limitations by learning a Koopman latent representation which allows us to infer symmetries of the system’s underlying dynamic." The statement is still unclear to me and most of the evaluations have been on same env (i.e. learned policy tested on the env in which the dataset was collected)
2. I am not sure how KFC++ is different from KFC

**Summary Of The Paper:**

The paper uses koopman theory to design principled data augmentation method for offline reinforcement learning. They learn a VAE style encoder-decoder model such that $D(E(s))=s$ and a forward model such that $F(s_t, a_t) = D((K_0 + \sum_{i=1}^m K_i, a_{t,i})E(s_t)) = s_{t+1}$. The koopman operator $K$ is then used to generate symmetries which are applied to both $s_t$ and $s_{t+1}$ as data augmentation during bellman error minimization. The resulting algorithm KFC and KFC++ leads to overall better data augmentation and improves for S4RL and CQL.

**Summary Of The Review:**

Weighing the strengths and concerns, I am recommending weak accept for now.

---

> ### Author Response · Authors · 2021-11-15
> **Response to Reviewer VgB7**
>
>
> We would like express our gratitude for your constructive feedback.
> Regrading your main objection on the out-of-distribution generalisation let us address your points
>
> * *"The statement is still unclear to me and most of the evaluations have been on same env (i.e. learned policy tested on the env in which the dataset was collected"*
> \
> This is correct. We do not test out-of-distribution (OOD) generalisation directly. The theorems guarantee however that theoretically  - and for systems obeying the  assumptions - symmetry transformations will map in principle to real new data. As this data is mostly OOD and the agent is trained on the latter it is evident that it will perform better on that data i.e. on OOD than agents who only have access to the original static dataset.
> \
> That said there are a variety of practical limitations which we stress in the paper, so the new data constitutes an approximation to real trajectories. So the question is if our symmetries maps are better suited than a model of comparable complexity (i.e. a neural net) modeling the data distribution? To which the answer is that our evidence suggests that they are.
> \
> Note that our symmetries map solutions for constant actions of the underlying system of ODEs into each other.  Thus for trajectories with complex action we find  a neighbourhood of the original data points in which we can reliable sample those trajectory information. Thus our symmetry map in combination with a data set containing meaningful trajectories is superior to a simple forward prediction model. Our ablation study in appendix C.3. proves this point.
> \
> Moreover, our framework can still be applied if the environment  fails to meet the theoretical assumptions. This follows form our main results in combination with ablation study in appendix C.1. In that case the symmetries are of an approximate system of ODEs which model the data distribution in which are learned in a self-supervised fashion by our approach.
> \
> Concludingly,  the above mentioned ablation studies  show a superiority of using symmetries to generate  out-of-distubtion data points. We are currently working on additional ablations studies on this point.
>
> **Minor**
>
> * *"I am not sure how KFC++ is different from KFC".*
>  \
>  Algorithm (I) in eq.\,(19) is KFC and (II) is KFC++. We have changed equation (19) in the revision to reflect this directly.

---

### Official Review · Reviewer_HCMe · 2021-11-08

**Correctness:** 4
**Technical Novelty And Significance:** 3
**Empirical Novelty And Significance:** 3
**Recommendation:** 6
**Confidence:** 4

**Main Review:**

I have a few questions that I believe the authors should address in their revised version to help the reader understand the method and its effectiveness.

1. Can you explain whether the Koopman representation in (12) holds for systems that are not control-affine?
2. The square is missing in the Bellman error in (1), (4), and (18).
3. Can you explain why using the symmetry generating function in (18) is not equivalent to using a different, say non-quadratic, Bellman error? In other words, I see KFC as a method that imposes the Bellman error at states that are outside the training dataset (this is great). But since we want hat{Q}_i and Q_i to be consistent across the entire state space in order to compute the correct fixed point. Therefore, the new states in (18) can be arbitrary and do not have to come from the symmetry generating function so long as they look like legitimate transitions of the dynamical system. For example, one can compute a PCA of the local neighborhood of the samples state and use the eigenvectors to perturb the states in the dataset and constrain the Bellman error to be small there. The paper does not convince that sampling states from the Koopman operator is essential.
4. Have you tried a standard VAE to sample new states, i.e., obtaining transitions (s_t, tilde{s_{t+1}}) where tilde{s_{t+1}} is the reconstruction of an interpolation of the latent vector corresponding to s_t and s_{t+1}? I believe this should be competitive.
5. In the same vein, are the transformed state trajectories sensible? For instance, in (16) if one computes the new trajectory as tilde{s_t), tilde{s_{t+1}), tilde{s_{t+2}}, ..., this trajectory could be replayed in the simulator or one can compute the actions/likelihood of the actions of the learned policy for this new state trajectory to ascertain that the transformations sampled from the VAE are sensible. This is essential in order to understand the method further because the symmetry in this computation can also be trivial because (I suspect) changing the origin of the state-space is a symmetry for all environments in Table 1.
6. This paper should study the proposed method better and conduct ablation studies. For instance, the cummulative reward of KFC should degrade as more samples are drawn from the Koopman VAE. How many extra samples are drawn to augment the existing offline datasets?
7. Do KFC and KFC++ use the same hyper-parameters as those in CQL? The narrative in Section 4 does not seem to clarify this. If not, it is difficult to claim that the improvements in performance in Table 1 come from the new method.
8. How many seeds were used to run the experiments in Table 1? Please provide p-values for whether KFC obtains a better cummulative reward than a baseline method.
9. The authors of https://arxiv.org/abs/2102.09225 found that the numerical performance reported in the original CQL paper is very different from the results obtained by implementing the code of the original authors. You should cite this paper and mention whether the numbers your observations with CQL (and KFC) are consistent with those of these authors.

**Summary Of The Paper:**

This paper develops a new method (Koopman Forward (Conservative) Q-learning, KFC in short) for offline reinforcement learning by extending the static dataset to include new states that are computed by learning an action-invariant Koopman latent representation of the system. The Koopman transformation to obtain the transformed state transitions is build using a variational auto-encoder. Experimental results are provided on D4RL, RoboSuite and MetaWorld benchmarks. This is a well-written paper (there are some minor typos which the authors should rectify). The method developed here shows improved performance over current methods in a number of benchmark problems.

**Summary Of The Review:**

This paper proposes an interesting idea based on Koopman theory that augments the limited dataset in offline RL but it is not clear from the current manuscript whether the performance of the method comes from the proposed innovations. The authors should present results of ablation studies that convince the reader that Koopman-operator is essential and a similar performance boost cannot be achieved by other augmentation methods (I have suggested a few above).

---

> ### Author Response · Authors · 2021-11-15
> **Response to Reviewer HCMe**
>
> We would like express our gratitude for your critical, constructive and inspiring feedback.
> In particular, it encouraged us to extend our ablation study on the proposed setup of a VAE with a state interpolation.
>
>  *   **3 and 4**  *"...Therefore, the new states in (18) can be arbitrary and do not have to come from the symmetry generating function so long as they look like legitimate transitions of the dynamical system... "*.
> \
> We agree with this statement entirely on practical grounds and both your suggestions may be efficient in producing legitimate state transitions to some extend.
>      However, on theoretical grounds symmetries of a dynamical system inherent way to describe local behavior and shift it by a larger distance in the state space.
>      Thus in principle one can sample new data for large shifts of the original data-distribution for certain symmetry transformations. To extract the latter has several practical limitations of course. We see our work as a starting point to introduce this new idea into the community. As we mentioned before our implementation of the model (VAE) and algorithm to extract symmetries is very simple, e.g. to use a simple MLP  for future state prediction is rather outdated.
>      Also note that one can view  our method as a new data-analysis tool which helps to understand symmetry based system transitions which may be of use in various other contexts.
> \
> Note that our symmetries map solutions for constant actions of the underlying system of ODEs into each other.  Thus for trajectories with complex action we find  a neighbourhood of the original data points in which we can reliable sample those trajectory information. Thus our symmetry map in combination with a data set containing meaningful trajectories is superior to a simple forward prediction model. Our ablation study in appendix C.3. proves this point.
>     Moreover, our framework can still be applied if the environment  fails to meet the theoretical assumptions. This follows from our main results in combination with the ablation study in appendix C.1.
> \
> Concludingly, we agree with you that a more quantitative comparison of our method to a prediction model is is very desirable and thus will find its way in our rebuttal version. We believe a prediction model (i.e. a neural net as in appendix C.3) constitutes the best comparison as the model complexity is identical, as well as the training procedure.  However, we find the idea of interpolating data-points in latent space of a VAE very interesting. We will try to incorporate it in one way ore another.
>   * **5:**
>      \
> Trajectories for changing actions are not connected by the same symmetry transformation (the proof is currently unpublished.) So this suggestion does not work on fundamental grounds.
>      Regarding your guess that our  symmetry transformations are trivial. Symmetries for control systems are in general not as simple as systems without control. Preliminary analysis  Walker-2D shows that the shifts follow complex meaningful patterns.
> \
> Moreover, the fact that an operator acting on the Koopman space commutes with the Koopman operator guarantees that it maps solutions  of the underlying system of differential equations to each other. In the non-control setting this simple translates to a mapping of solutions with different initial conditions. It is a symmetry in that regard. So the question one could ask is the following: How do we know that the Koopman operator in our simple VAE model is a good approximation to the real dynamics and the real Koopman operator of the system? To which the answer is one does not in general. So what one practically does is to find a Koopman operator which models the data distribution i.e. a system of coupled linear ODEs. The computed symmetries are of the latter approximate system which are learned in a self-supervised fashion. We are working on further ablation studies discussing the non-triviality of the learned symmetry maps.
> * **6:**   We sample only from augmented data.
> * **7:**  Thank you for pointing this out; all base hyper-parameters are shared between the baseline CQL model and KFC-variants. We will clarify this in the final version.
> * **8:**  We run each experiment for 5 trial each and simply report the mean, which is in accordance to similar papers in the community such as D4RL, CQL, BRAC, S4RL.
> * **9:**   Training the baseline CQL model took some effort, but eventually, with the help of the authors of the CQL paper, we were able to get similar performance to the paper. To ensure consistent benchmarking, we used the same numbers as reported in the base papers for both CQL and BRAC.
>
> **Minor:**
>
> *  **1:** Your questions regards to non-affine control systems, right? Eq.(12) may hold for non-affine systems in certain cases but not in general. We will extend the introduction to Koopman theory in the revised version to present Eq.(12) in a bit more detail.
> *  **2:**  Thanks for pointing this out.

---

### Author Response · Authors · 2021-11-15
**General Response to all Reviewers**


We would like to express our sincere gratitude to the reviewers for their  critical, constructive and inspiring assessment of our work.
To address the common concerns we are currently working on several modifications and are updating the draft in accordance to the changes suggested.
Those regard the following major points

* Firstly, an extension of our ablation study "Koopman-prediction" addressing the question of the relevance of the symmetries to generate new data points rather than a model oriented approach.
*  Secondly, a quantitative analysis of the symmetry transformations learned in  self-supervised fashion by our framework. Arguing in favor for their non-triviality.
* Thirdly, a mathematically more precise theory section regarding definitions and assumptions building the basis of our theoretical contribution.


Regarding the first point above. The question if other methods may be used to sample legitimate data-points  - of which several suggestions were made by the reviewers - is in our opinion best done justice by an extension of the already existing ablation study "Koopman-prediction" in appendix (C.3). The results in C.3 are that a prediction model falls behind in almost all categories of Mujoco at D4RL which is a strong indicator for the correctness of a superiority of the use of symmetries.

\
**A note on the simplicity of the current algorithm:** Let us stress a crucial point to the reviewers which has not been emphasized enough in our work. Algorithmically the symmetry maps are derived in two distinct ways  **KFC** and **KFC++** i.e. case (I)  and (II) in eq.(19), respectively . The latter, constitute a simple starting point to extract symmetries from our setup. More elaborate studies employing the extended literate on Koopman spectral analysis are desirable. Moreover, it is desirable to extend our framework to more complex latent space descriptions such as e.g. world models. Both on a theoretical as well as a practical level. It is our opinion that by doing so there is significant room for improvement both in the accuracy of the derived symmetry transformations as well their induced  performance gains of Q-learning algorithms.
 Note that currently our VAE model is of very simple nature and the symmetries are extracted in a rather uneducated way. While the Sylvester algorithm simply converges to one out of many symmetry transformations  for the   KFC++ algorithm we omit all the information of the imaginary part, let alone utilize concrete spectral information.



 Please refer to our individual responses to the reviewers for more detailed explanations.

---

### Author Response · Authors · 2021-11-23
**Rebuttal version main changes**

Thank you for your patience while we were updating our rebuttal version.
We are glad to see broad consensus among the reviewers leaning towards acceptance. **We would like to thank [Reviewer SsNL](https://openreview.net/forum?id=q1QmAqT_4Zh&noteId=XaXBL1oGz6L) for the score update.**

We address any remaining common concerns through several significant modifications in the manuscript:

* Firstly, an extension of our ablation study "Koopman-prediction" in appendix (D.4) addressing the question of the relevance of the symmetries to generate new data points rather than a model oriented approach. We evaluate it on the entire D4RL benchmark set and find that compared to a  predictive model the symmetry shifts are superior in terms of performance.

* Secondly, we have revised section (3.1) to provide a mathematically more precise introduction into symmetries and ODEs, where we introduce the concept of a local Lie Symmetry Group. We also express the assumptions building the basis of our theoretic contribution more clearly. In particular, we establish a relation in Theorem 3.5  to those symmetry groups with an extension of the proofs in the appendix (B).  (Please note that due to the additional material some other theoretical parts of the work are now shifted to the appendix (A). Also Theorem 3.7 previously is now Theorem 3.5 etc.)
\
Moreover, to provide a better transition between the theoretical and the empirical part of our work we rebuild section (3.2) and added some material in that regard.

* Thirdly, in appendix (D.3) we provide a quantitative analysis of the symmetry transformations learned in  self-supervised fashion by our framework.  Arguing in favor for their non-triviality over a random shift in the VAE's latent space. We perform an active (online) evaluation in the Mujoco environment of the random and symmetry shifted states, respectively.

---

### Decision · Program_Chairs · 2022-01-20

**Decision:**

Reject

**Comment:**

This paper proposes to improve offline RL by a data augmentation technique that exploits the symmetry of the dynamics using Koopman operator. The idea is interesting but the draft at its current form has several weaknesses as pointed out by the reviewers. The scores are borderlines at this point. I read the paper and find myself agree with reviewer ohJ3 in both the lack of  clarity and the gap in theory and empirical results. The math presentation still  a careful check and improvement. Eq(1)-(4) are already fairly confusing (should $Q_i$ and $\pi_i$ be replaced by $Q$ and $\pi$ in Eq(1)-(4), and $\hat Q$ by $\hat Q_i$ in Eq (2)-(3)?). I would like to suggest the authors to add a self-contained algorithm box for the practical algorithm procedure. Do the readers really need to understand the full Koopman theory (section 3.1) before understanding the algorithm? The authors could think about if it is better to present the practical algorithm first with minimum math, and then analyze the property of the algorithm using the math tools (and in this case, make it clear what theoretical guarantees we get exactly). I think making the paper more accessible can help the paper gain more popularity in ML readers.